# MAGELLAN: Metacognitive predictions of learning progress guide autotelic LLM agents in large goal spaces

Loris Gaven [1]    Thomas Carta [1]    Clément Romac [1 2]    Cédric Colas [1 3]    Sylvain Lamprier [4]    Olivier Sigaud [5]
Pierre-Yves Oudeyer [1]

## Abstract

Open-ended learning agents must efficiently prioritize goals in vast possibility spaces, focusing on those that maximize learning progress (LP). When such autotelic exploration is achieved by LLM agents trained with online RL in high-dimensional and evolving goal spaces, a key challenge for LP prediction is modeling one's own competence, a form of metacognitive monitoring. Traditional approaches either require extensive sampling or rely on brittle expert-defined goal groupings. We introduce MAGELLAN, a metacognitive framework that lets LLM agents learn to predict their competence and LP online. By capturing semantic relationships between goals, MAGELLAN enables sample-efficient LP estimation and dynamic adaptation to evolving goal spaces through generalization. In an interactive learning environment, we show that MAGELLAN improves LP prediction efficiency and goal prioritization, being the only method allowing the agent to fully master a large and evolving goal space. These results demonstrate how augmenting LLM agents with a metacognitive ability for LP predictions can effectively scale curriculum learning to open-ended goal spaces.

## 1. Introduction

Humans are open-ended learners, continuously exploring and developing new skills through their lifetime. A key mechanism to enable this remarkable capacity is *curiosity-driven learning* — the intrinsic motivation to explore for

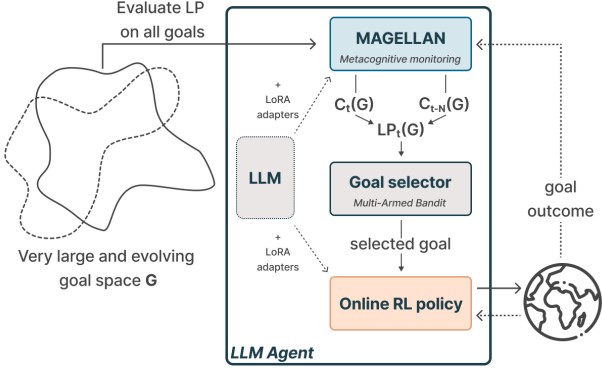

*Figure 1.* **Navigating large goal spaces with MAGELLAN**: During training, our LLM agent uses MAGELLAN to estimate its past and current competence to compute absolute learning progress (ALP) on each goal. Given the per-goal ALP, the LLM agent's goal selector chooses the next goal to practice proportionally to their ALP. The LLM agent then performs a trajectory to achieve this goal and the outcome is used to update both the LLM agent with online RL and MAGELLAN's competence estimation.

the sake of learning and discovery (Berlyne, 1954; Kidd & Hayden, 2015). Crucially, humans are *autotelic learners* intrinsically motivated to represent, invent, select and solve their own goals (Colas et al., 2022a). To navigate in a possibly infinite space of goals, without time to explore it exhaustively, they are equipped with intrinsic motivation signals (Baldassarre & Mirolli, 2012; Gottlieb & Oudeyer, 2018). Research on human curiosity has shown the key role of one such intrinsic signal: Learning Progress (LP), i.e. improvement of one's own ability to solve goals (Kaplan & Oudeyer, 2007). Computational modeling work showed both how it enables efficient automatic curriculum learning (Lopes & Oudeyer, 2012a; Poli et al., 2024b) and how it generates developmental trajectories that simulate key properties in the development of human infants (Oudeyer & Smith, 2016). Recently, several experimental paradigms where humans were free to explore various learning activities confirmed that humans use metacognitive LP monitoring to explore and prioritize goals (Ten et al., 2021; Leonard et al., 2023; Sayalı et al., 2023; Poli et al., 2024a).

[1]Inria (Flowers), University of Bordeaux, France [2]Hugging Face [3]MIT, Computational Cognitive Science Lab, Cambridge, MA, USA [4]Univ Angers, LERIA, SFR MATHSTIC, F-49000 Angers, France [5]Sorbonne Université, ISIR, Paris, France. Correspondence to: Loris Gaven <loris.gaven@inria.fr>.

*Proceedings of the 42$^{nd}$ International Conference on Machine Learning*, Vancouver, Canada. PMLR 267, 2025. Copyright 2025 by the author(s).

Inspired by open-ended learning in humans and other natural systems, research in AI and artificial life has studied how to build machines with similar capabilities (Schmidhuber, 2013; Jiang et al., 2023; Sigaud et al., 2023). A promising direction focuses on developing autotelic artificial agents that, like humans, can self-generate their learning curriculum by progressively exploring goals with maximum LP (Baranes & Oudeyer, 2013; Colas et al., 2022b). This approach efficiently allocates the agent's learning time by avoiding goals that are either too easy or too difficult (Portelas et al., 2020b; Romac et al., 2021), enabling even physical robots to acquire complex skills like tool use in just a few dozen hours (Forestier et al., 2022). However, while these methods show promise in constrained settings, scaling them to open-ended learning remains challenging. The key difficulty lies in efficiently estimating an agent's current competence and expected LP across potentially infinite, evolving, and high-dimensional goal spaces — a fundamental challenge we address in this paper.

Parallel to this, a recent breakthrough has emerged in training large language model (LLM) agents to learn goal-directed behaviors through online interaction with their environment (Carta et al., 2023; Wen et al., 2024a;b). These LLM agents possess a critical capability for open-ended learning: they can leverage the structure of language to generalize effectively, transferring skills learned from practiced goals to semantically similar goals. Using language-instructed agents also offers extensive expressiveness for specifying goals, as required for the open-ended learning setting. This usually induces huge goal spaces, for which prioritization is crucial. However, current approaches leveraging LP for this prioritization fall short at handling such discrete, high-dimensional and structured goal space. They either work only on small low-dimensional goal spaces (Baranes & Oudeyer, 2013; Portelas et al., 2020a; Kanitscheider et al., 2021; Zhang et al., 2024) or rely on expert-defined goal groupings to reduce the number of goals (Colas et al., 2019; Akakzia et al., 2021; Kumar et al., 2024). In particular, none of them are able to capture the semantic relationships between goals to efficiently estimate an LLM agent's generalization abilities.

In this paper, we study how to estimate LP over natural language goals such that an LLM agent learning with online RL in an interactive environment could increase its overall competence as efficiently as possible. For this, we introduce **MAGELLAN**, for **M**et**A**cognitive **GE**neralization of **L**earning progress in **LAN**guage model agents. MAGELLAN leverages the LLM inside the agent to learn an LP estimator that automatically learns semantic relationships and tracks competence transfer between goals in a sample efficient manner (see Figure 1). We evaluate MAGELLAN in the Little-Zoo environment specifically designed as a carefully controlled experimental setup for commonsense-

based generalization of agents in a textual environment. In particular, we study the following scientific questions:

• **Q1.** Given an initial set of language goals, how does MAGELLAN's estimation of a learner's competence compare to more classic methods? How does this estimation scale with the size of the goal space?

• **Q2.** Can MAGELLAN be used by an online RL LLM agent to self-organize an efficient learning curriculum over these goals?

• **Q3.** How well can MAGELLAN's estimation generalize to predict the agent's competence on unseen goals?

• **Q4.** When these new unseen goals are introduced throughout training, can MAGELLAN leverage its generalization abilities to integrate new goals into the curriculum seamlessly?

We show that MAGELLAN 1) accurately and efficiently approximates LP, 2) allows an LLM agent to master all goals from Little-Zoo while prior methods fail when not provided extensive expert knowledge, and 3) generalizes its LP estimation to never-seen goals, enabling faster adaptation to evolving goal spaces. Moreover, we show MAGELLAN learns to cluster goals and achieves results comparable to an LP estimator with expert-defined groups.

## 2. Related Work

### 2.1. Goal selection in autotelic agents

Autotelic agents exploring vast goal spaces face a critical challenge: they must prioritize which goals to pursue to efficiently develop general competence (Colas et al., 2022b). The automatic curriculum learning (ACL) community has developed various approaches to address this challenge (Portelas et al., 2020b), leveraging different forms of intrinsic motivation: pursuing goals of intermediate difficulty (Held et al., 2017; Racaniere et al., 2020; Castanet et al., 2023), seeking novelty or uncertainty (Warde-Farley et al., 2019; Pong et al., 2019; Pitis et al., 2020), maximizing regret w.r.t. an optimal policy's performance — mostly used in Unsupervised Environment Design (UED) methods (Dennis et al., 2020; Jiang et al., 2021) — or maximizing learning progress (LP) (Stout & Barto, 2010; Matiisen et al., 2020; Fournier et al., 2018; Portelas et al., 2020a; Colas et al., 2019; Kanitscheider et al., 2021; Kovač et al., 2023; Zhang et al., 2024). ACL methods also differ in their use of the goal space: while some methods compute metrics over the goal space and sample from it (e.g. (Matiisen et al., 2020; Kanitscheider et al., 2021; Portelas et al., 2020a)), others generate goals (e.g. (Held et al., 2017; Castanet et al., 2023; Dennis et al., 2020)). LP-based methods have proven particularly robust, especially upon a limited training budget in

which it is not possible to learn all goals (Lopes & Oudeyer, 2012b). They adapt to the agent's capabilities *without requiring environment knowledge* (e.g. which is necessary when using regret) and avoid common pitfalls like getting stuck on goals where progress plateaus or chasing uncontrollable novelty. The key challenge with LP approaches lies in efficiently estimating progress over large goal spaces, which is the focus of our work.

## 2.2. Computing LP over goals

Learning Progress (LP) measures the expected future improvement in achieving a goal through practice (Oudeyer et al., 2007). Since future progress cannot be directly measured, most approaches use past progress as a proxy, with the recent exception of (Kumar et al., 2024)'s Bayesian prediction model. The most direct approach to estimate LP is to regularly reevaluate the agent's competence for each goal (Kanitscheider et al., 2021; Zhang et al., 2024), which accurately captures competence transfer — the phenomenon where practicing one goal affects performance on other goals. However, this becomes computationally prohibitive for large discrete goal spaces and is just impossible for continuous ones. One way to address this is to only rely on online estimations, where a goal's estimated competence is only updated when this goal is practiced. Online estimations nonetheless fail to capture competence transfer and existing methods addressed this by grouping goals with similar competence together. For continuous spaces, approaches either learn to partition the space directly when dimensionality is low (Oudeyer et al., 2007; Baranes & Oudeyer, 2013; Portelas et al., 2020a), or first embed high-dimensional goals into a lower-dimensional space before partitioning (Laversanne-Finot et al., 2018; Kovač et al., 2023). For discrete spaces, methods typically rely on expert-defined groupings (Stout & Barto, 2010; Matiisen et al., 2020). However, these grouping approaches are inherently brittle: they assume no transfer between groups while potentially masking competence variations within groups. This limitation is particularly acute for high-dimensional structured spaces like natural language, where competence transfer naturally occurs between semantically similar goals regardless of predefined groupings. Instead, MAGELLAN leverages an LLM's semantic understanding to dynamically model competence transfer between goals, enabling efficient and adaptive LP estimation without requiring predefined groupings or exhaustive evaluation.

## 2.3. Autonomous LLM agents

Recent work has explored using Large Language Models (LLMs) to solve complex tasks in interactive environments. Early approaches focused on direct action prediction using LLMs, either incorporating environmental feedback (Huang et al., 2022; Yao et al., 2022; Hao et al., 2023; Shinn et al., 2023; Wang et al., 2024) or operating without it (Ahn et al.,

2022). However, these methods did not update the LLM's knowledge through environmental interactions. A new direction emerged with GLAM (Carta et al., 2023), followed by (Wen et al., 2024b;a; Zhou et al., 2024) that ground LLMs in interactive environments using online RL. The resulting LLM agents demonstrated remarkable generalization across language tasks; however, they lack the autotelic mechanisms necessary for navigating expansive goal spaces. In this paper, we enhance a SAC-GLAM (Gaven et al., 2024) LLM agent with metacognitive abilities, enabling it to estimate its LP and prioritize goals within large language spaces

## 3. Methods

In this section, we detail how MAGELLAN learns a metacognitive module that estimates and generalizes an agent's LP over language goal spaces. We then explain classic LP baselines against which MAGELLAN is compared. Finally, we introduce the Little-Zoo environment, specifically designed to study commonsense-based generalization abilities of LLM agents when facing large language goal space.

### 3.1. Problem statement

Let $\mathcal{M} = (S, A, \mathcal{T}, R)$ be an MDP, with $S$ a set of states, $\mathcal{T}$ the transition function, $A$ the action space and $R$ the reward function. Let $G$ be a goal space and $\Pi$ the policy space. We define a competence function $C_{\mathcal{M}, \pi} : G \to \mathbb{R}$ that indicates the competence of a policy $\pi \in \Pi$ for a goal in $\mathcal{M}$.[1] The final aim is to find the optimal policy $\pi^*$ that maximizes

$$J_{\mathcal{M}}(\pi) = \mathbb{E}_{g \sim \mathcal{U}(G)}[C_{\mathcal{M}, \pi}(g)],$$

where $U(X)$ is the uniform distribution over a set $X$.

In this paper, we focus on episodic online goal-conditioned RL with sparse and binary rewards, defined on a goal-augmented MDP $(S, A, \mathcal{T}, G, R)$, with $R : S \times G \to \{0; 1\}$ a binary success function indicating whether a state $s$ satisfies a goal $g$. Here, we define $G = \{S_0 \times I | S_0 \subseteq S\}$, with $I$ an instruction space and $S_0$ the set of initial states. We consider a textual environment where a prompting function $\phi : S \times I \to \mathcal{V}^K$ is given to transform any pair (state, instruction) into a textual prompt of $K$ tokens in a given vocabulary $\mathcal{V}$. Thus, from the agent side, the policy $\pi$ selects any action $a_h \in A$ by sampling from a categorical distribution $\pi(.|\phi(s_h, i))$ at any step $h$ of the episode.

For our competence function we use the success probability $\mathbb{P}_\pi(s_0, i)$ defined as the probability for $\pi$, starting from $s_0$, to fulfill $i$ within $H$ steps: $\mathbb{P}_\pi(s_0, i) = \mathbb{E}_{\tau \sim \pi(\tau | \phi(s_0, i))}[r_{\tau, i}]$, with $r_{\tau, i} = \mathbb{1}(\exists s_h \in \tau, R(s_h, i) = 1)$ the goal outcome

---

[1]In all generality a competence function does not assume the goal to be inside the MDP. For instance, $g$ could ask for a maximum number of steps.

of episode $\tau$ for instruction $i$, $\mathbb{1}$ the indicator function and $\pi(\tau|\phi(s_0, i))$ the distribution of episodes of $H$ steps induced by $\pi$ to fulfilling $i$ from $s_0$. In this setting our objective becomes:

$$J(\pi) = \mathbb{E}_{s_0 \sim \mathcal{U}(S_0), i \sim \mathcal{U}(I)} \left[ \mathbb{P}_\pi(s_0, i) \right].$$

However, given the possibly huge number of goals $(s_0, i)$, the direct maximization of the problem becomes particularly inefficient. Our aim is to leverage transfer of competence between goals and focus during training on the ones maximizing LP. We denote as $\pi^t$ the policy obtained after $t$ episodes using an RL algorithm, with $\Gamma^t$ the set of all trajectories collected during training using $\{\pi^{k-1}|k \in [|1, t|]\}$. The goal of each episode is sampled using a task selector $\eta_G$ that selects a goal based on collected trajectories $\eta_G(\Gamma^t) = g$. Given a budget of $T$ training episodes, we thus consider the problem of approaching the optimal selector $\eta_G^* = \arg\max_{\eta_G} J(\pi^T)$. In particular, we build on prior work to construct $\eta_G$ on a proxy of the LP at each training episode $t$. We define the LP for any goal $g = (s_0, i)$ as the improvement of the policy at episode $t$ after $k$ episodes on goal $g$: $LP_{\pi^t}^k(g) = \mathbb{P}_{\pi^{t+k}}(g) - \mathbb{P}_{\pi^t}(g)$. As highlighted in Section 2.2, since computing the future competence on all goals is intractable, prior approaches approximate future progress with past progress ($LP_{\pi^t}^k(g) \approx \mathbb{P}_{\pi^t}(g) - \mathbb{P}_{\pi^{t-k}}(g)$). Nonetheless, accurately estimating past and current competence remains a challenge in large discrete goal spaces.

### 3.2. Metacognitive generalization of learning progress in language model agents

With MAGELLAN, we propose to learn estimators of the current and past policy's competence for any goal. As opposed to prior works, which either consider all goals independently or use goal groupings, we argue that learning goal-conditioned estimators would allow generalization between similar goals without defining any clear group. We propose to leverage the LLM used by our agent to learn the parameters $\theta_t$ of a competence estimator $C_{\theta_t}(g)$ for a policy $\pi_t$ on a goal $g$. We compute $C_{\theta_t}(g)$ by giving $g$ in the LLM's prompt, which produces a latent representation on top of its final decoder block for the last token. We use a Multi-Layer Perceptron (MLP) to output the estimated competence based on this representation. We train both the LLM and the MLP, leveraging the LLM's ability to project goals into a latent space where semantically similar goals are close. By updating the estimated competence of one goal, this allows MAGELLAN to also update close goals.

In practice, we maintain a buffer $\mathcal{D}_t$ which contains, for the $M$ most recent training episodes at $t$ (i.e., $\tau^{t-M} \dots \tau^t$), their corresponding pair of goal and outcome (i.e. $(g = (s_0, i), r_{\tau, i})$ for each $\tau$). As this work

focuses on success probability (i.e., we want $C_{\theta_t}(g) \approx \mathbb{P}_{\pi^t}(g)$), we train $C_{\theta_t}$ using stochastic gradient descent to minimize the binary cross-entropy: $\mathcal{L}(\theta_t) = \mathbb{E}_{(g,r)\sim\mathcal{D}_t} \left[ BCE(r, C_{\theta_t}(g)) \right]$.

We maintain another buffer $\mathcal{B}_t$ storing the last $N$ weights of our competence estimator: $\mathcal{B}_t = [\theta_{t-N}, \theta_{t+1-N}, \dots, \theta_t]$. Weights are added to the buffer every time the competence estimator is updated, enabling access to estimations of the policy's competence from time $t$ to $t - N$. Using this information, we estimate the absolute LP (ALP) Baranes & Oudeyer (2013); Kanitscheider et al. (2021), tracking both progress and forgetting, as follows:

$$A\hat{L}P_{\pi_t}(g) = |C_{\theta_t}(g) - C_{\theta_{t-N}}(g)|. \tag{1}$$

This ALP estimation can subsequently be used to structure the agent's curriculum. We apply the multi-armed bandit goal selection scheme introduced by (Lopes & Oudeyer, 2012b) where each arm is a goal, and its utility is MAGELLAN's estimate of the ALP of this goal. Goals are then sampled proportionally to their estimated ALP with an annealing $\epsilon$-greedy scheme ($\epsilon$ decreasing from 1 to 0.2). In practice, we train two separate versions of the same initial LLM (using LoRA adapters (Hu et al., 2022)): one for the policy and one for MAGELLAN's current competence estimator. We show in Appendix D.1 ablations on architectural choices indicating that 1) keeping the LLM frozen leads to poor results, highlighting the need for a dynamic representation space (see also Figure 6), and 2) training separate LoRA adapters for the policy and MAGELLAN leads to more stability.

### 3.3. Classic ALP baselines

Following the literature on ALP in Section 2.2, we implement classic approaches, focusing on two dimensions. First, we consider Online (Baranes & Oudeyer, 2013; Matiisen et al., 2020) vs Evaluation-based ALP (Kanitscheider et al., 2021; Zhang et al., 2024) estimation. Then, we consider directly using the goal space (Portelas et al., 2020a; Kanitscheider et al., 2021) or using expert-defined groups of goals with assumed competence transfer (Stout & Barto, 2010; Colas et al., 2019). The latter requires extensive expert knowledge (EK) given the absence of automatic approach for discrete goal spaces. As expert-defined groups are created beforehand, no competence transfer is assumed across groups, which is likely to happen in spaces like natural language, where transfer occurs between semantically close goals regardless of groups.

We thus implement four baselines (see all details in Appendix C.3):

- **Eval-ALP**: Every $N$ episodes, training stops and the

agent is separately evaluated on each goal to obtain a competence estimate. The per-goal ALP is the absolute difference between estimates at $t$ and $t - N$. The same goal selection scheme as in MAGELLAN is used according to the per-goal ALP estimations.

- **EK-Eval-ALP**: Every $N$ episodes, training stops and the agent is evaluated on multiple goals randomly sampled in each expert-defined group to obtain a per-group averaged competence. The per-group ALP is computed using the absolute difference between the competence at $t$ and $t - N$. The goal selection process first selects a group using the same selection scheme as MAGELLAN for goals. Given a selected group, a goal from this group is randomly selected.
- **Online-ALP**: At each goal practiced, the observed policy's competence is added to a buffer of $2M$ past experiences for this goal. The ALP is computed using the absolute difference between the average competence over the last and first $M$ experiences in the buffer. The same goal selection scheme as MAGELLAN and Eval-ALP is used.
- **EK-Online-ALP**: At each goal practiced, the observed policy's competence is added to a buffer of $2M$ past experiences for the goal's expert-defined group. The per-group ALP is computed using the absolute difference between the average competence over the last and first $M$ experiences in the group's buffer. The same goal selection scheme as EK-Eval-ALP is used.

We summarize the four methods above and MAGELLAN in Table 1 based on their *Efficiency* (i.e. computational cost introduced by additional evaluations), *Competence Transfer tracking* and *no Expert Knowledge requirement*. We consider a method's efficiency as "high" if it does not require any additional evaluation (i.e. it only uses the performance observed on goals sampled), and as "low" otherwise. We evaluate the competence transfer tracking using the following criteria:

- absence of +: the estimated competence is updated only on sampled goals.
- +: the estimated competence is updated on a predefined group the sampled goal belongs to.
- ++: the estimated competence is updated on a dynamically learned group the sampled goal belongs to.
- +++: the estimated competence is updated on all goals.

We provide in Appendix B the same table for all prior works covered in Section 2.2.

### 3.4. The Little-Zoo environment as a testbed

Evaluating commonsense-based generalization of LLM agents in textual environments requires several key properties. The environment must be fully text-based, with all observations, actions, and goals expressed in natural lan-

|  | Eff. | Transf. | No EK |
|---|---|---|---|
| **EK-Eval-ALP** | low | + | × |
| **Eval-ALP** | low | +++ | ✓ |
| **EK-Online-ALP** | high | + | × |
| **Online-ALP** | high |  | ✓ |
| **MAGELLAN** | high | +++ | ✓ |

*Table 1.* Comparison of ALP estimation methods. We use the following dimensions: computational Efficiency, competence Transfer tracking, and required Expert Knowledge.

guage. It should feature a diverse set of goals with varying difficulty levels, enabling the assessment of the agent's ability to learn and generalize complex skills. Additionally, these goals should be organized into hidden families based on commonsense knowledge, allowing for targeted evaluation of generalization capabilities.

Existing environments for LLM agents do not have such requirements. Creative environments like Minecraft (Johnson et al., 2016) or Crafter (Hafner, 2022) rely on image-based observations and require an image captioner to use LLM agents. Textual environments like BabyAI-text (Carta et al., 2023) focus on navigation skills without any commonsense-based generalization. Although WordCraft (Jiang et al., 2020) incorporates commonsense-based goals, no relationship between goals exists, limiting the analysis of the agent's generalization abilities. To address these gaps, we introduce Little-Zoo, a novel environment explicitly designed to meet these criteria.

Built upon the Playground environment (Colas et al., 2020), Little-Zoo is fully text-based, with observations, goals, and actions expressed in natural language. It features objects that can be combined together and are grouped into the following hidden categories: furniture (which cannot be combined), plants, herbivores, and carnivores. Given the set of all objects and a set of instructions, Little-Zoo's goal space is the combination of all possible instructions and scene initializations. The feasibility of a goal thus depends on the objects available, making most combinations infeasible and not trivial to detect (see Figure 2 and Appendix A.3). For instance, these are respectively feasible and infeasible goals: *"Goal: Grow deer. You see: baby deer, bookshelf, water, tomato seed."*; *"Goal: Grow deer. You see: baby deer, bookshelf, baby lion, tomato seed."* (water is missing in the last one). Instructions are hierarchically structured, ranging from simple grasping tasks to more complex sequences involving object interactions (e.g. "growing" animals). The complete goal space contains approximately 20 million combinations. In our experiments, we subsample goals with the following proportions: 80% of the goals are impossible 16% involve grasping, 3.2% involve growing plants, 0.7% involve herbivores, and 0.1% involve carnivores. These proportions correspond to proportions in the complete goal

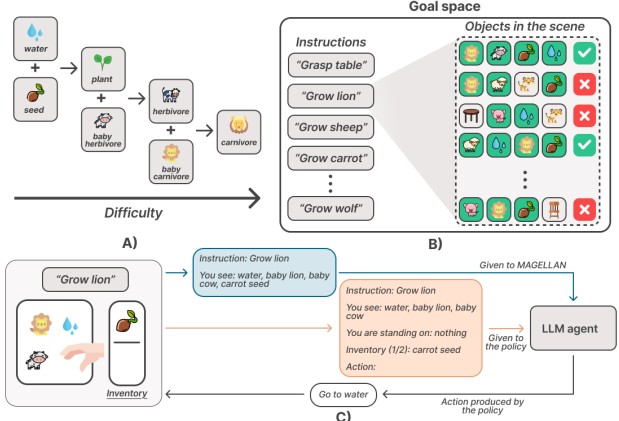

*Figure 2.* A) Little-Zoo's tech tree. B) Little-Zoo's goal space is composed of all the possible combinations between instructions and objects that can be in the scene. Most object configurations make an instruction infeasible (e.g. "grow lion" is impossible with the second configuration, as water, needed to obtain plants, is missing). C) Little-Zoo provides a textual description that is given in our LLM agent's prompt.

space (see Appendix A.4).

Little-Zoo is a deterministic, fully-observable and episodic environment: the agent begins an episode by standing on nothing, with full visibility of the whole scene. The action space consists of 8 actions, including movement to objects, grasping, and releasing objects. Observations include the objects in the scene, the ones in the agent's inventory, as well as the object the agent is standing on. See Appendix A for details on Little-Zoo.

## 4. Experiments

We provide empirical answers to our scientific questions using experiments with 8 different random seeds in the Little-Zoo environment. For our LLM agent, we use SAC-GLAM (Gaven et al., 2024) to finetune Flan-T5 248M (Raffel et al., 2020) as per SAC-GLAM's experiments. We compare MAGELLAN to the classic approaches presented in 3.3. For methods accessing external expert knowledge, we use Little-Zoo's hidden goal families (grasp any object, grow plant, grow herbivore, grow carnivore) but add only possible goals in these groups. Indeed, these baselines are based on groups predefined in advance by human experts with a strong assumption: the goals within a group share the same learning dynamics and therefore the agent's competence is the same over all goals in the group. *If impossible goals were included, these groups would lose their relevance.* Moreover, because of the large number of impossible goals, the average competence within each group will always be very close to 0. There will be no progress niche that the method can use to generate a curriculum, and

performance will probably be very close to the random baseline. We thus provide an additional group containing all impossible goals. In all our experiments, we use success rate (i.e. average outcome over multiple trials for a goal) noted SR as the observed competence.

We first study how well MAGELLAN's competence estimation compares to baselines (**Q1**). Then, we study how the different methods (except Eval-ALP and EK-Eval-ALP, which are too costly to run while EK-Online-ALP provides a good estimation of their performance) compare when scaffolding the LLM agent's curriculum (**Q2**). We show how these competence estimators also generalize to goals not seen during training (**Q3**). Finally, we study how all methods adapt as the goal space evolves (**Q4**) by replacing all goals at different points throughout training.

### 4.1. How well does MAGELLAN estimate competence (Q1)

To assess the ALP methods' ability to efficiently estimate competence, we designed an experimental setup in which our LLM agent was trained for 50k episodes on the Little-Zoo environment with varying goal space sizes (25k, 50k, 100k), while keeping the same repartition between goal types. As computing the expected ALP to train this agent is intractable, one could argue that Eval-ALP is the best approximation. However, it remains too computationally costly to run, even when performing only 50k training episodes with 25k goals. We thus chose to sample goals according to EK-Eval-ALP's estimations. To obtain an accurate estimate, we perform 2048 per-group evaluations every 1000 episodes. The per-group competence evaluated by EK-Eval-ALP is consequently our competence reference, and we compare the other methods against it. For Eval-ALP, we consider it to have zero error, and its computational cost can be estimated without running it. For MAGELLAN and Online-ALP, we average the per-goal competence over groups to compute the error w.r.t. EK-Eval-ALP. Figure 3 shows the average error on competence throughout training along and the cost of competence evaluation (i.e. the total number of episodes used only to evaluate competence).

As indicated in Table 1, MAGELLAN performs on a par with Eval-ALP, showing that it accurately estimates the transfer of competence while using online estimations. We also observe similar competence errors to methods using expert-defined groups, hinting at MAGELLAN's abilities at learning semantic relationships between goals. We provide a more in-depth analysis of such relationships in Section D.4. Finally, MAGELLAN achieves this performance without an estimation cost.

To demonstrate that our method generalizes beyond the Little-Zoo environment, we conducted an additional experiment using goals derived from the OpenR1-Math-220k

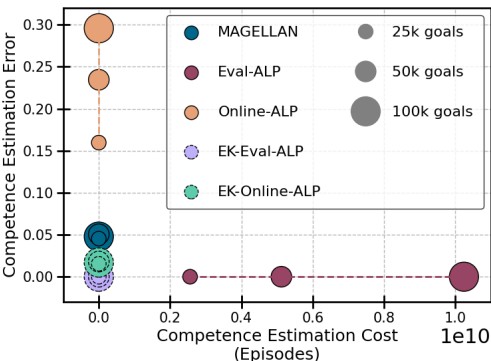

*Figure 3.* Scaling of competence estimation error and competence estimation cost (i.e. total number of additional evaluation episodes) when increasing the goal space size.

dataset (Face, 2025). In this setup, rather than training a real agent, we simulate the learning of an agent that progressively acquires skills in three categories: *Algebra*, then *Geometry*, and finally *Number Theory*. We compare the competence estimation from MAGELLAN and Online-ALP to the underlying true competence. Figure 4 shows that MAGELLAN accurately tracks the agent's competence and distinguishes between the different mathematical categories, clearly outperforming Online-ALP. Additional results, including experiments in the BabyAI-Text environment (Chevalier-Boisvert et al., 2019; Carta et al., 2023) and an ablation study on the impact of LLM size within MAGELLAN, are provided in Appendix D.2.

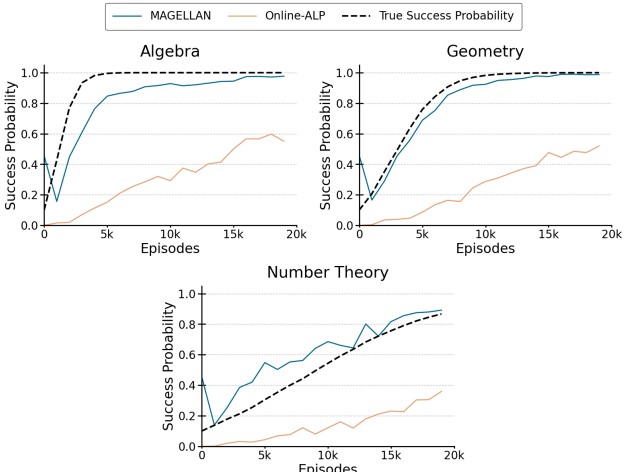

*Figure 4.* Competence estimation on OpenR1-Math-220k. MAGELLAN (blue) accurately tracks competence across Algebra, Geometry, and Number Theory, closely matching true success probabilities and outperforming Online-ALP (orange).

## 4.2. Training an LLM agent with MAGELLAN (Q2)

As demonstrated in 4.1, MAGELLAN provides a superior competence estimation than Online-ALP. We further investigate whether this improvement translates into a better curriculum and improved overall goal mastery. We train our LLM agent on the goal space of Little-Zoo with 25k goals for 500k episodes using four methods: MAGELLAN, Online-ALP, EK-Online-ALP and "Uniform", where goals are sampled uniformly. We do not report EK-Eval-ALP, as we report EK-Online-ALP which produces similar competence estimation with no cost. We report the agent's SR every 5000 training episodes by evaluating it on 64 goals uniformly sampled for each category. Figure 5 shows the evolution of SR averaged over all categories. Our results show that MAGELLAN is the only method without expert-defined grouping to obtain an SR of at least 90% in all categories. It also masters the categories significantly faster than baselines. Despite MAGELLAN's similar competence estimation as EK-Online-ALP, the latter learns faster by leveraging the expert-defined groups to better explore. This is because MAGELLAN explores by uniformly sampling goals whereas EK-Online-ALP uniformly samples groups, easily discarding impossible goals.

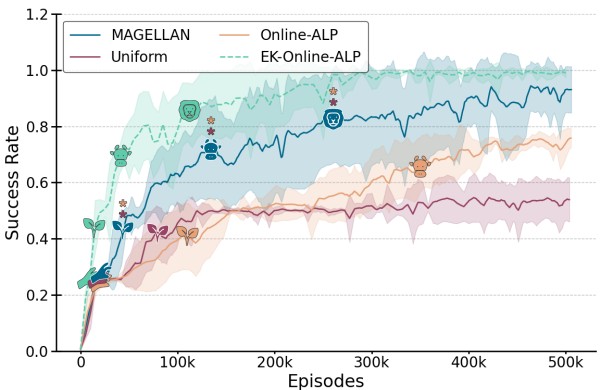

*Figure 5.* Evolution of the observed competence (SR) when evaluating policies on 64 training goals per category every 5000 episodes. We report the average SR over evaluated goals along with standard deviation (8 seeds). Icons indicate the average time step at which a method mastered a goal (i.e. SR > 90%). We add stars to MAGELLAN, denoting significantly earlier mastery of a category compared to the method with the star's color (p-value $< 8 \times 10^{-4}$). The dotted line (EK-Online-ALP) indicates that the method relies on expert knowledge.

## 4.3. MAGELLAN's generalization abilities (Q3)

We move further and study the generalization abilities of both our LLM agent and competence estimators. While Section 4.2 evaluates each policy on 64 goals per category that belong to the training goal space, this section reports

evaluation on a held-out test set composed of unseen goals. As in the previous section, evaluations were performed every 5000 training episodes. However, instead of reporting the policy's observed competence (SR) during evaluation, we report the difference between the observed competence and the competence estimated by the policy's ALP method. We show in Table 2 the average error over training.

By only tracking competence on goals practiced by the policy (i.e. the ones from the training goal space), Online-ALP cannot provide any estimation for new unseen goals and uses its default competence of 0. This leads to the largest error among methods except for "Grow carnivore" goals as the policies trained with Online-ALP never mastered these goals (see Figure 5). MAGELLAN successfully generalizes its competence estimation and obtains a small error. Finally, EK-Online-ALP produces accurate estimations based on expert knowledge of which group each test goal belongs to. Appendix D.4 provides detailed results indicating our LLM agents do not perfectly generalize, which explains MAGELLAN and EK-Eval-ALP estimation error.

*Table 2.* We evaluate the policies trained with each ALP method on a held-out test set. We show the difference between the observed and predicted competence. Online-ALP's performance on "Grow carnivore" is simply explained by the fact that its policies never mastered this goal category.

| Categories | MAGELLAN (Mean ± Std) | Online-ALP (Mean ± Std) | EK-Online-ALP (Mean ± Std) |
|---|---|---|---|
| **Grasp** | **0.01 ± 0.00** | 0.98 ± 0.00 | 0.01 ± 0.00 |
| **Grow plant** | **0.05 ± 0.03** | 0.78 ± 0.07 | 0.03 ± 0.01 |
| **Grow herbivore** | **0.08 ± 0.05** | 0.34 ± 0.18 | 0.06 ± 0.02 |
| **Grow carnivore** | 0.30 ± 0.16 | **0.00 ± 0.00** | 0.28 ± 0.08 |
| **Mean** | **0.11 ± 0.06** | 0.53 ± 0.06 | 0.09 ± 0.03 |

We further investigate MAGELLAN's generalization abilities by projecting train (Q2) and test (Q3) goals from a single seed into the LLM embedding space MAGELLAN used. The embedding space is plotted both before and after training (Figure 6) with projections obtained via t-SNE (van der Maaten & Hinton, 2008). The initial embedding space lacks any discernible structure for goal classification. Post-training, the space exhibits significant restructuring, with similar goals accurately clustered. A small subset of "Grow carnivore" goals are misclassified as impossible, likely due to incomplete mastery of this category by the LLM agent.

Furthermore, the spatial arrangement of goals correlates with estimated success rates: newly learned goals tend to lie near the boundary with impossible goals. Additionally, goals from the test set are well clustered, demonstrating strong generalization. We show in Appendix D.4.2 that MAGELLAN also makes different clusters for impossible goals based on the infeasibility reason.

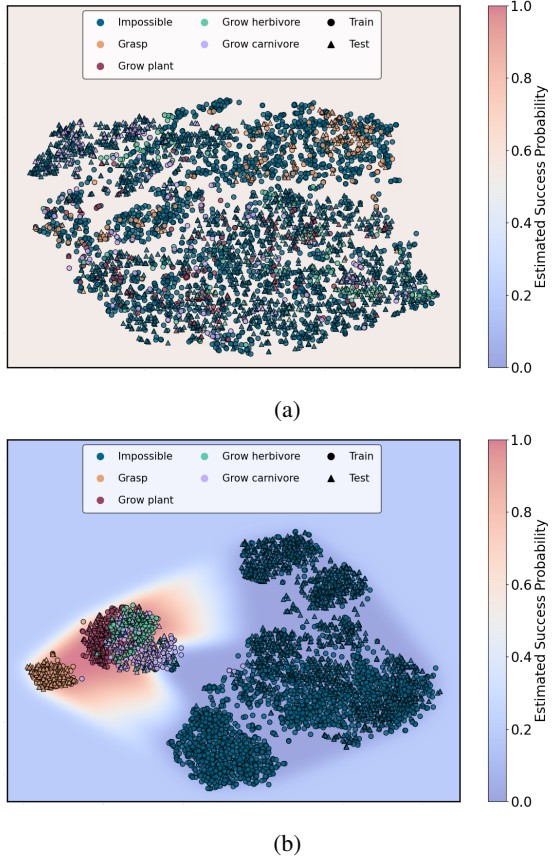

(a)

(b)

*Figure 6.* MAGELLAN's LLM embedding space displayed using t-SNE with goals used in Q2 (Train) and Q3 (Test), along with the estimated success probability and linear interpolation between goals. We show the embedding space for a single seed (a) before training and (b) at the end of the 500k training steps. We see that impossible goals have been left aside, and that the other goals with a high estimated success probability are clustered consistently.

### 4.4. MAGELLAN's adaptation to evolving goal spaces (Q4)

Finally, we investigate how each ALP method can adapt when the goal space evolves. For this, we isolate the training of one seed using MAGELLAN in Section D.3. Every 50k episodes over the 500k training episodes, we stop training, replace the training goals with the ones in our held-out test set, and start four trainings (with 8 seeds each) with this new goal space: one with each ALP method for 50k steps. We expect MAGELLAN to quickly adapt to new goals leveraging semantic relationships between the new and old goals. Online-ALP starts with a competence estimation of 0 on the new goals as in Section 4.3. For EK-Online-ALP, up to the episode where we change the goal space, we make it track the policy's competence in parallel to MAGELLAN. It thus starts with a per-group competence estimation when the goals are replaced along with the information of which

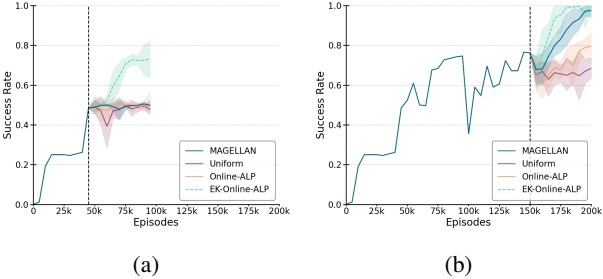

(a)                    (b)

*Figure 7.* **Adaptation tests**: Using a single's seed training of 500k episodes, we stop and replace all goals with unseen ones every 50k episodes. We then resume training and sample goals using each method for 50k training episodes. We show two isolated and representative points of goal replacement: (a) there is no ALP on any goal (after 50k training episodes), and (b) some goals (here, "Grow carnivores" after 150k training episodes) have a high ALP. We report the evolution of SR when evaluating the policies on 64 goals per category from the new training set every 5000 episodes. Results show the average competence over evaluated goals along with standard deviation (8 seeds).

expert-defined group each new goal belongs to. We study all adaptation training in Appendix D.5 but isolate and study in Figure 7 two of them chosen as representative of the scenarios encountered:

- **Scenario zero LP (Figure 7a):** The agent has mastered the "Grasp" and "Grow plants" goals and has 0 ALP across all goals. In this scenario, all ALP estimations are equivalent. EK-Online-LP manages to discover new ALP niches faster as all impossible goals are in the same group.
- **Scenario high LP (Figure 7b):** The agent is getting a high ALP as it is learning some "Grow carnivores" goals. Here, MAGELLAN outperforms baselines by generalizing its ALP estimation and continuing training on these goals. MAGELLAN even gets on par performance with EK-Online-LP.

## 5. Conclusion

In this paper, we introduce MAGELLAN, a metacognitive module designed to efficiently estimate an LLM agent's learning progress (LP) across large language-defined goal spaces. By leveraging the LLM within the agent, MAGELLAN accurately and efficiently estimates LP by learning semantic relationships between goals. This approach fundamentally differs from prior methods, which either struggle to scale to large goal spaces or rely on expert-defined goal categories. Using MAGELLAN's LP estimations, the LLM agent can effectively structure its curriculum, enabling it to master all goals within an extensive goal space. In contrast, previous methods achieve only partial mastery in the absence of expert-defined categories. Moreover, MAGEL-

LAN's ability to capture semantic relationships between goals allows it to assess the agent's competence on unseen goals and rapidly adapt as the goal space evolves. While our study provides an in-depth analysis of MAGELLAN's capabilities in a controlled textual environment, the method itself is highly generalizable. It offers a goal prioritization strategy applicable to any learner operating in a large, high-dimensional, structured, and discrete goal space. Notably, goal spaces involving code appear particularly promising, given the proficiency of current LLMs in code generation (Wang et al., 2024; Pourcel et al., 2024). Additionally, traditional automatic curriculum learning settings—where goals are not language-defined—could also benefit from MAGELLAN's ability to uncover relationships between goals. Beyond artificial learners, MAGELLAN may also prove valuable for human learning, particularly in educational domains where learners must navigate a large space of language-defined problems, such as mathematical word problems (Doroudi et al., 2019). Classical LP measures have already been shown to enhance personalized curricula in educational technologies (Clement et al., 2015), suggesting MAGELLAN's potential impact in this area.

## Impact Statement

Large Language Models (LLMs) are now widely used in real-world applications, and the recent rise of LLM agents empowered with actions such as web search or tool use greatly enhanced their capabilities. In this work, we study open-ended learning mechanisms for such LLM agents, paving the way for agents solving very large and diverse task spaces. While our results indicate significant improvement for LLM agents to learn in such task spaces, our experiments were limited to small-scale LLMs and well-controlled experimental testbeds. Consequently, we do not recommend generalizing our findings to real-world open-ended learning settings. Additionally, our work introduces an efficient method for measuring Learning Progress (LP), an intrinsic motivation signal notably used for prioritizing tasks given to human learners in multiple real-world educational technologies. However, our experiments did not include human learners, and the real impact of our method on accurately measuring human LP remains to be evaluated.

## Acknowledgment

This work was granted access to the HPC resources of IDRIS under the allocation A0171011996 made by GENCI. We acknowledge funding from the European Commission's Horizon Europe Framework Programme under grant agreement No 101070381 (PILLAR-robots project), the French Defense Innovation Agency, and the ANR AI Chair grant ANR-19-CHIA-0004. We also thank Nicolas Yax for kindly proofreading the paper and providing valuable feedback.

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

# Appendices

This supplementary material provides additional results, discussion, and implementation details.

- Section A details our Little-Zoo environment.
  - Section A.1 explains the environment mechanics and how the different actions affect the environment. It also gives more details about optimal trajectories.
  - Section A.2 details the goal space generation.
  - Section A.3 explains the different types of impossible goals. This section also gives examples for each type of impossible goal.
  - Section A.4 contains the goal space's distribution per goal type.

- Section B extends the table from Section 3.3, comparing the previous LP estimation methods.

- Section C contains the implementation details of MAGELLAN and the baselines.
  - Section C.1 explains the LLM-based RL agent used in our experiments, the associated hyperparameters and the prompt used.
  - Section C.2 details MAGELLAN architecture, hyperparameters and the prompt used.
  - Section C.3 contains the additional details for implementing the baselines.
  - Section C.4 is about the compute budget.

- Section D extends the results given in the Section 4 on the experiments of the paper.
  - Section D.1 is an ablation on MAGELLAN's architecture.
  - Section D.2 develops the results given in Section 4.1 with plots for goal spaces of different size.
  - Section D.3 contains several additional results for Section 4.2.
    * Section D.3.1 gives the evolution of the success rate per method and per goal type.
    * Section D.3.2 is an analysis of the goal sampling strategy of each method.
  - Section D.4 develops several aspects of MAGELLAN's generalization abilities from Section 4.3.
    * Section D.4.1 specifically examines the ability to generalize success probability estimations to test goals across different goal types.
    * Section D.4.2 describes the embedding evolution during training.
    * Section D.4.3 conducts a detailed analysis of the embeddings of impossible goals.
  - Section D.5 expends on the results found in Section 4.4.
    * Section D.5.1 contains all the curves from the experiment about the adaptation to an evolving goal space.
    * Section D.5.2 provides a quantitative analysis of this experiment.

# A. Little-Zoo environment

## A.1. Environment mechanics

There are 5 categories of elements the agent can interact with: the objects (i.e. "bookshelf"), the water, the plants (i.e. tomato) which can bee `seed` and `grown`, the herbivores (i.e cow) which can be `baby` or `grown`, and the carnivores (i.e. lion) which can be `baby` or `grown`.

To interact with any element of the environment, multiple actions are accessible. First, the agent can use `"Go to {element}"` to move to any element. You can only move to an element (not a random point in the plane) and after a "Go to" action you are placed on the element. When performing the `"Grasp"` action, the element the agent is standing on is added to its inventory (up to 2 items). Some elements can interact with each other. To make two elements interact, the agent must release one element in its inventory by using the `"Release {element}"` action while standing on the other element. Interactions between three elements are also possible: the agent has to hold two elements in its inventory and perform `"Release all"` while standing on the third element. The "Release" action is only accessible if an interaction can be made between the element to release and the one the agent is standing on. Therefore, the agent must carefully manage its inventory to avoid filling it with useless elements without being able to release them. At each turn you get a full description of the environment with `"You see {element_1}, {element_2}, {element_3}, {element_4}"`.

To grow a plant, an herbivore or a carnivore, the agent needs to follow a certain sequence of actions:

- To `"Grow a plant"`, the agent has to `"Release water"` while `"Standing"` on the plant seed.

- To `"Grow an herbivore"`, the agent has to `"Release a grown plant"` while standing on a baby herbivore.

- To `"Grow a carnivore"`, the agent has to `"Release a grown herbivore"` while standing on a baby carnivore.

Table 3 summarizes the optimal action strategy for each category. In order to maintain a coherent level of difficulty between the different categories, we fix the maximal number of actions allowed to the agent to solve a goal as $150\%$ of the minimal number of actions required to solve the goal. The required exploration to discover strategies has, therefore, the same level of difficulty for all goals.

Table 3. The optimal trajectories per categories.

| Categories | Optimal action strategy | Minimal number of actions | Maximal number of actions |
|---|---|---|---|
| Grasp X | Go to X, Grasp | 2 | 3 |
| Grow plant | Go to water, Grasp, Go to plant, Release water | 4 | 6 |
| Grow herbivore | Go to water, Grasp, Go to plant, Release water, Grasp, Go to herbivore, release plant | 7 | 11 |
| Grow carnivore | Go to water, Grasp, Go to plant, Release water, Grasp, Go to herbivore, release plant, Grasp, Go to carnivore, release | 10 | 15 |

## A.2. Goal space generation

To define our goal space, we start from the sets of objects $\mathcal{O}$, plants $\mathcal{P}$, herbivores $\mathcal{H}$, carnivores $\mathcal{C}$ and the element water (each set being of size 6). One goal $g$ in the goal space is defined using 5 elements $(E_0, E_1, E_2, E_3, E_4) \in \{\mathcal{O}, \mathcal{P}, \mathcal{H}, \mathcal{C}, \text{water}\}$ and by using the objective function $\mathcal{F}_g$ from the set $\{\text{Grasp}\{X\}, \text{Grow}\{X\}\}$ on $E_0$. Thus, we get the goal $g = (\mathcal{F}_g(E_0), E_1, E_2, E_3, E_4)$. As each set contains 6 elements, we get a total of $19,531,250$ goals. The vast majority of these goals are impossible for various reasons (see Appendix A.3).

In our experiments, we subsample two goal spaces: a train and held-out test space. To generate them, we sample the total number of impossible goals $n_{impo}$ we want to have, then sample among the possible goals: $\frac{n_{impo}}{5}$ goals where $E_0 \in \mathcal{O}$,

$\frac{n_{impo}}{5^2}$ goals where $E_0 \in \mathcal{P}$, $\frac{n_{impo}}{5^3}$ goals where $E_0 \in \mathcal{H}$, and $\frac{n_{impo}}{5^4}$ goals where $E_0 \in \mathcal{C}$. Appendix A.4 gives more details on the goal repartition.

### A.3. Example of impossible goals

In our environment, a goal is defined as a combination of elements present in the environment and an instruction, such as "grasp the rabbit". Some combinations are incompatible such as:

```
Grasp rabbit
You see:  bookshelf, baby lion, desk, baby cow,
```

where there is no rabbit. The generation of our environment leads to having $80\%$ of impossible goals in the goal space, which is similar to any complex environment (Zhang et al., 2024; Matthews et al., 2024). However, conversely to (Zhang et al., 2024), identifying impossible goals and avoiding to select them is not trivial, as there are multiple reasons why a goal is impossible.

The easiest reason for impossibility is the absence of the element the agent has to act on:

- Absence of the element to grasp
  ```
  Goal:  Grasp bed
  You see:  bookshelf, baby lion, desk, baby cow.
  ```

- Absence of the element to grow
  ```
  Goal:  Grow deer
  You see:  baby giraffe, bookshelf, water, tomato seed.
  ```

Another reason is that the element cannot follow the dynamic required in the objective:

- Growing an element that is neither an animal nor a plant
  ```
  Goal:  Grow desk
  You see:  bed, baby bobcat, baby elephant, door.
  ```

A more difficult reason that requires a better understanding of the underlying mechanisms of the environment is the lack of at least one element necessary to reach the objective grow:

- The lack of water:
  ```
  Goal:  Grow cucumber
  You see:  cucumber seed, carrot seed, baby coyote, baby wolf,
  Goal:  Grow coyote
  You see:  cucumber seed, berry seed, baby coyote, baby cow.
  ```

- The lack of a plant preventing the agent to grow an herbivore resulting in the impossibility to grow a carnivore:
  ```
  Goal:  Grow coyote
  You see:  water, baby elephant, baby coyote, baby cow.
  ```

Consequently, to efficiently build a curriculum, being able to understand why a goals is impossible and quickly generalize to infer the other impossible goals is crucial.

### A.4. Goal repartition

At the end of the goal space generation, we have 19531250 goals distributed as presented in Figure 8a. However, using all the goals was computationally impossible considering all the comparisons we made. Moreover, the repartition with $92\%$ of impossible goals and $7.53\%$ of "Grasp" goals made the epsilon-greedy exploration of our goal selector too slow to discover LP niches in an acceptable compute budget. Subsequently, we sample a training goal space and a testing goal space following the distribution given in Figure 8b. Both goal spaces are composed at $80\%$ of impossible goals, the remaining goal categories becoming less and less represented as the difficulty increases. This repartition synthetically simulates how natural language goals mostly lead to sequences of words without any meaning. This structure makes the strategy of uniformly sampling goals inefficient and underlines the necessity of a curriculum to be able to reach the hardest goals in the allocated training budget (500k episodes in our experiments).

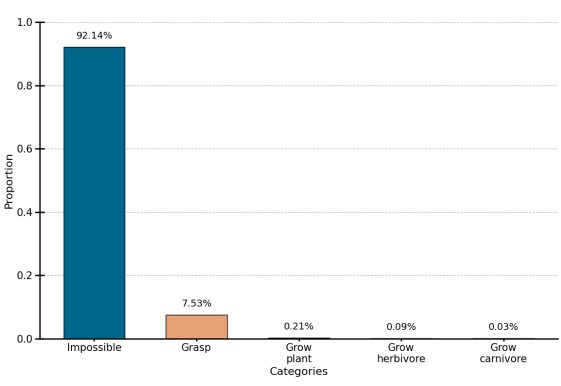

(a) Goal distribution per categories on the full goal space.

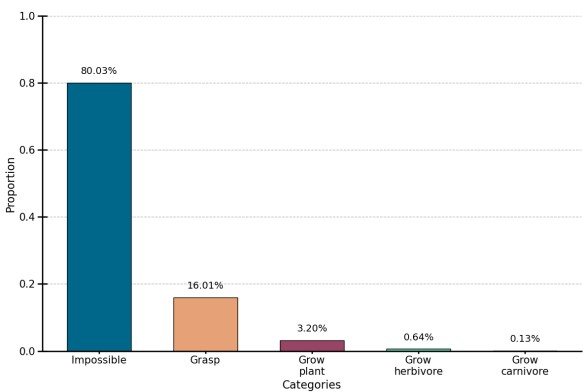

(b) Goal distribution per categories in our experiments.

*Figure 8.* We first generate the full goal space whom the distribution is given Figure 8a then for computational reasons we sample a smaller goal space with the following distribution Figure 8b.

## B. Comparison of LP methods

We compare prior work computing LP for automatic curriculum learning under the dimensions from Section D.2. We show the comparison in Table 4 evaluating methods:

- **Efficiency**: Computational cost of additional evaluation episodes not used to train the policy.

- **Competence transfer tracking**: How well does the method track all the possible competence transfer.

- **No expert knowledge required**: if they require any external expert knowledge such as pre-defined goal groupings.

We consider a method's efficiency as "high" if it does not require any additional evaluation (i.e. it only uses the performance observed on goals sampled), and as "low" otherwise. We evaluate the competence transfer tracking using the following criteria:

- absence of +: the estimated competence is updated only on sampled goals.
- +: the estimated competence is updated on a predefined goal subset the sampled goal belongs to.
- ++: the estimated competence is updated on a dynamically learned goal subset the sampled goal belongs to.
- +++: the estimated competence is updated on all goals.

| Methods | Efficiency | Competence transfer tracking | No expert knowledge required |
|---|---|---|---|
| (Oudeyer et al., 2007) | high | ++ | ✓ |
| (Baranes & Oudeyer, 2009) | high | ++ | ✓ |
| (Baranes & Oudeyer, 2013) | high | ++ | ✓ |
| (Moulin-Frier & Oudeyer, 2013) | high | ++ | ✓ |
| (Moulin-Frier et al., 2014) | high | ++ | ✓ |
| (Portelas et al., 2020a) | high | ++ | ✓ |
| (Forestier & Oudeyer, 2016) | high | ++ | ✓ |
| (Kovač et al., 2023) | high | ++ | ✓ |
| (Stout & Barto, 2010) | high | + | × |
| (Lopes & Oudeyer, 2012b) | high | + | × |
| (Matiisen et al., 2020) | high | + | × |
| (Fournier et al., 2018) | high | + | × |
| (Colas et al., 2019) | high | + | × |
| (Blaes et al., 2019) | high | + | × |
| (Akakzia et al., 2021) | high | + | × |
| (Kumar et al., 2024) | high | + | × |
| (Kanitscheider et al., 2021) | low | +++ | ✓ |
| (Zhang et al., 2024) | low | +++ | ✓ |

*Table 4.* Comparison of prior work w.r.t. their LP estimation approach. We use the following dimensions: computational efficiency, dynamical competence transfer assumptions and no required expert knowledge.

# C. Implementation details

To facilitate reproduction and future work, we provide our code at `https://github.com/flowersteam/MAGELLAN`.

## C.1. LLM-based RL agent

We use the SAC-GLAM method from (Gaven et al., 2024) with the hyperparameters listed in Table 5. Following SAC-GLAM, the Q-value head is a two-layer MLP with 1024 ReLU-activated units, applied to the last hidden state of the decoder. The policy and the critic share the same LoRA adapters. Additionally, we apply a warm-up phase of 10 updates, during which only the Q-function is trained.

*Table 5.* SAC hyperparameters

| Variable | Value |
|---|---|
| Update frequency | 64 |
| Number of updates | 1 |
| Batch size | 256 |
| Discount factor | 0.99 |
| Optimizer | Adam |
| Critic learning rate | $1 \times 10^{-4}$ |
| Actor learning rate | $1 \times 10^{-4}$ |
| Entropy coefficient | auto |
| Entropy coefficient initialization | 0.05 |
| Target entropy | $-0.125$ |
| Entropy coefficient learning rate | $1 \times 10^{-3}$ |
| n-step | 3 |
| Replay buffer capacity | 500000 |

The prompt used as the LLM agent's observation is shown in Prompt C.1.

> **Prompt C.1: RL Agent observation**
>
> Goal: Grow lion
> You see: water, carrot seed, baby lion, baby cow
> You are standing on: nothing
> Inventory (0/2): empty
> Action:

## C.2. MAGELLAN

Our competence estimator uses a single-layer MLP with 128 units and Tanh activations, applied to the last hidden state of the LLM. It is updated every 32 policy updates, with batch sampling that prioritizes recent data. The sampling distribution is determined by $\frac{i}{\sum_{j=1}^{M} j}$, where $i$ represents the position of a data point in the buffer $\mathcal{D}$ (with 1 being the oldest and $M$ the buffer size). This strategy ensures that the competence predictor remains responsive while preserving batch diversity, incorporating data from different versions of the learning agent to help mitigate noise introduced by the fluctuations in the RL agent's learning process.

To compute the LP, we store the weights of our competence estimator in the buffer $\mathcal{B}$. This is done by saving the LoRA adapters and the MLP weights whenever the competence estimator is updated. The window for LP computation is defined by $|\mathcal{B}| \times$ *update frequency*. The LP is calculated by comparing the competence estimation obtained with the oldest and most recent weights in the buffer.

The MAGELLAN hyperparameters are provided in Table 6.

*Table 6.* MAGELLAN hyperparameters

| Variable | Value |
|---|---|
| $\epsilon$ start | 1 |
| $\epsilon$ end | 0.2 |
| $\epsilon$ decay period | 320 |
| $\mathcal{B}$ size | 100 |
| $\mathcal{D}$ size | 5000 |
| Batch size | 256 |
| Optimizer | Adam |
| Learning rate | $1 \times 10^{-4}$ |
| Update frequency | 32 |

The prompt given to the competence estimator is the same as the prompt given to the RL agent Prompt C.2.

> **Prompt C.2: MAGELLAN prompt**
>
> Goal: Grow lion
> You see: water, carrot seed, baby lion, baby cow
> You are standing on: nothing
> Inventory (0/2): empty
> Action:

## C.3. Baselines

We compared our method against four baselines: Online-ALP, Eval-ALP, EK-Online-ALP, and EK-Eval-ALP. Below, we outline the implementation details for the different families of methods:

- **Non-EK methods** For methods that do not rely on expert knowledge, competence and LP estimations are computed individually for each goal.

- **EK methods** For methods incorporating expert knowledge, goals are grouped into five buckets, with competence and LP computed at the bucket level.

- **Online methods** In online methods, a buffer of size $M$ stores goal-outcome pairs. The agent's current competence is estimated from the most recent half of the buffer, while past competence is derived from the oldest half. Non-EK methods maintain a separate buffer for each goal, whereas EK methods use a buffer for each bucket.

- **Eval methods** In evaluation-based methods, the agent's competence is assessed every $N$ episodes. For non-EK methods, the agent is evaluated $k$ times on each individual goal, while for EK methods, evaluations are performed $k$ times per bucket.

The hyperparameters for Online methods are presented in Table 7, while those for Eval methods are shown in Table 8.

*Table 7.* Online methods hyperparameters

| Variable | Value |
| --- | --- |
| $\epsilon$ start | 1 |
| $\epsilon$ end | 0.2 |
| $\epsilon$ decay period | 320 |
| Buffer size | 100 |

*Table 8.* Eval methods hyperparameters

| Variable | Value |
| --- | --- |
| $\epsilon$ start | 1 |
| $\epsilon$ end | 0.2 |
| $\epsilon$ decay period | 320 |
| Eval frequency | 1000 |
| Number of eval | 2048 |

### C.4. Compute budget

To optimize GPU VRAM usage during training, we employ 4-bit quantization techniques as described in (Dettmers et al., 2023). We use a vectorized version of Little-Zoo with 32 instances of the environment running (synchronously) in parallel. In order to accelerate training of our LLM agent (both its policy with online RL and MAGELLAN), we leverage Lamorel[2] to deploy 2 instances of the LLM in parallel. Thus, we distribute both the forward passes to compute actions' probability or LP and training in a Data Parallelism setting. When using Flan-T5 250M, each LLM instance is distributed (Vertical Model Parallelism) over one Nvidia H100 80GB GPUs requiring thus a total of 2 Nvidia H100 80GB GPUs to run an experiment (1 GPU $\times$ 2 LLM instances). For one seed of one ALP method trained in Section 4.2, performing 500k training episodes requires 80 GPU hours on the Nvidia H100 80GB.

---

[2]https://github.com/flowersteam/lamorel

# D. Additional results

## D.1. Ablations on the MAGELLAN architecture

In order to implement MAGELLAN, we try various architectures presented in Figure 9. In all of them, to reduce the computational cost, we froze the LLM and used LoRA adapters (Hu et al., 2022) for training the policy (along with its Q-value) and the SR estimator. The architectures tested are:

- **A: Different adapters for the policy and MAGELLAN** (Figure 9a). This is the architecture adopted in the paper. The environment representation for the policy and the goal space representation for the competence estimator are learned separately on different adapters.

- **B: Shared adapters for the policy and MAGELLAN**(Figure 9b). We test whether a shared useful representation between the policy and competence estimator can emerge through training.

- **C: Shared adapters but only trained with the policy loss**(Figure 9c). We use a single set of LoRA adapters to train both the policy and the competence estimation module, but apply the latter's gradient only to the MLP outputting the competence. This ablation tests if the policy loss on its own creates semantical relationships between goals. We argue that training of the policy should push towards a clustering of the goal space.

- **D: Adapter only for the policy, frozen LLM representation for MAGELLAN**(Figure 9d). In this ablation, we test if MAGELLAN can learn to predict competence using the LLM's original latent space. Learning to estimate competence over a fixed representation is closer to prior works such as ALP-GMM (Portelas et al., 2020a).

We compare the learning dynamics of the four architectures in Figure 10. We observe no difference in their ability at learning "grasp" goals. However, the agents with Architecture A continue to progress on other goal types whereas the ones using Architectures B, C, and D stagnate between 50k and 100k episodes before progressing again. At the end of training (500k episodes), agents using architecture A have learned all goal types. The ones based on Architectures B and C learned up to "grow herbivore" goal types. The agents that utilize Architecture D succeed plateaued on "grow plant".

To gain a finer comprehension of the learning dynamics implied by the different architectures, we look in Figure 11 at the embeddings at the end of training for each architecture.

Architecture A (Figure 11a) and architecture B (Figure 11b) shared very similar representations with in both cases goals clustered between "grasp", "grow" and impossible goals. Inside the cluster "grow" the different element types (i.e. plants, herbivores, carnivores) are also clustered. Nonetheless, as seen in Figure 10, Architecture B does not master "grow carnivore" goals and classified them as impossible. This indicates that obtaining a shared latent space for the policy and MAGELLAN is possible but slows down skill acquisition.

Architecture C (Figure 11c) where MAGELLAN uses the representation solely changed by the policy still manages to accurately estimate competence. However, predicting competence is much more difficult using this architecture as the goal space is not modified ease competence estimation. Yet, using only the policy gradient still improves the initial latent space of the LLM as shown by Architecture D (Figure 11d).

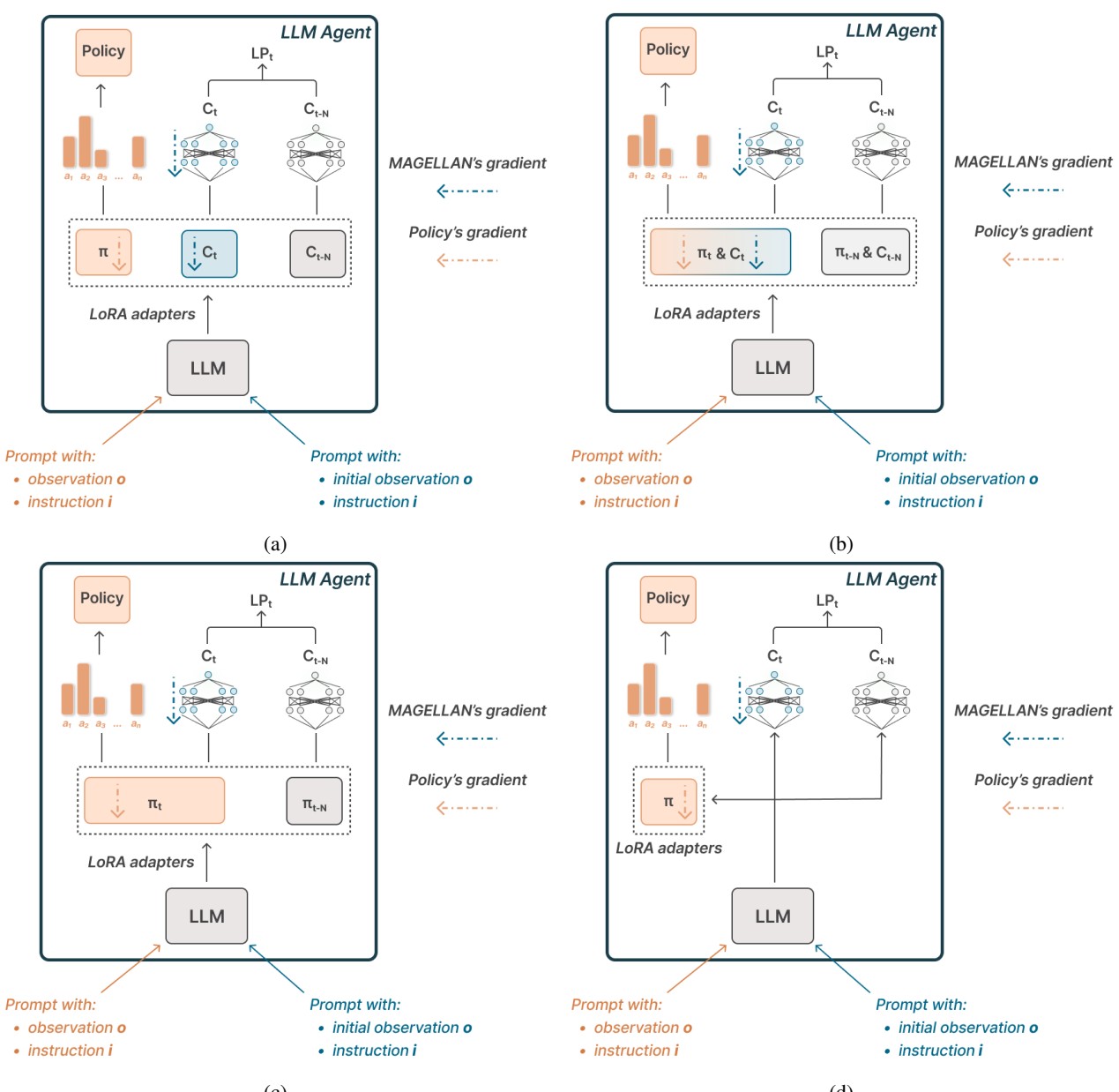

*Figure 9.* Different architectural choices in MAGELLAN: (a) we learn separate LoRA adapters between the policy and MAGELLAN (used in the paper); (b) we share adapters and update them using both the policy and MAGELLAN gradient; (c) we share adapters but they are only updated by the policy gradient; (d) MAGELLAN directly uses the latent representation produced by the pretrained LLM.

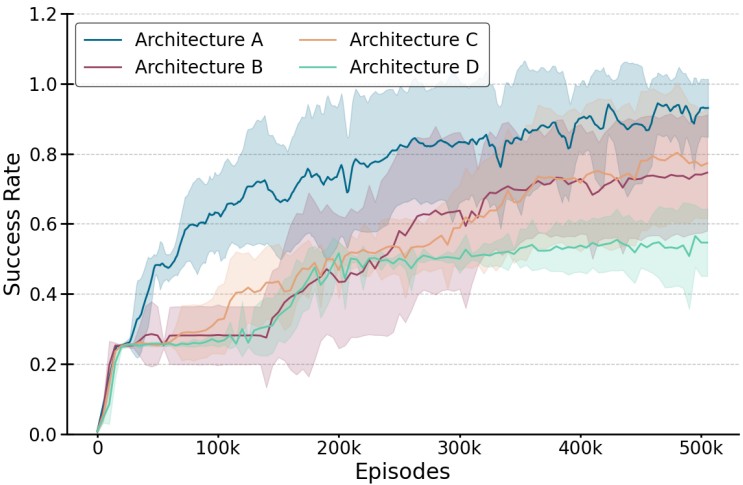

*Figure 10.* Training curves of the four different possible architecture for MAGELLAN. We use 8 seeds to plot the mean and the standard deviation (shadow area around the solid line).

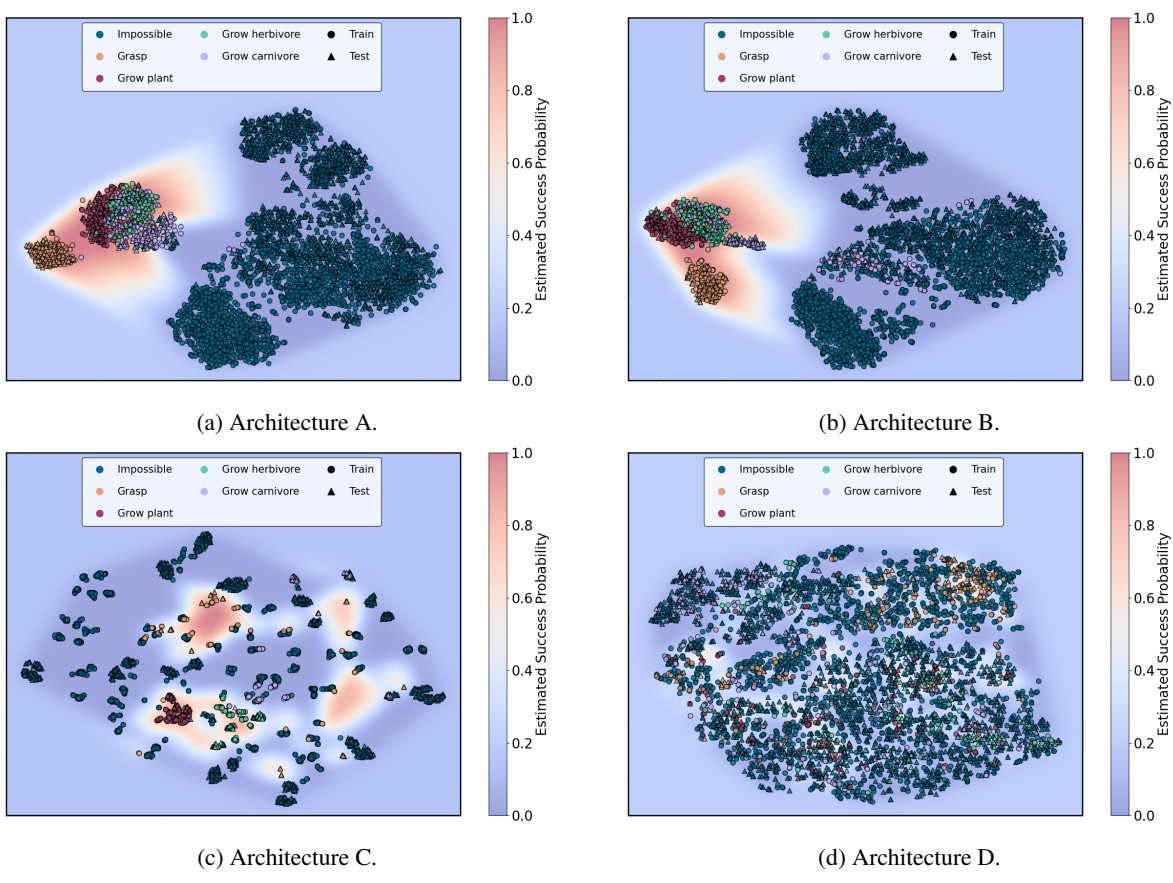

(a) Architecture A.

(b) Architecture B.

(c) Architecture C.

(d) Architecture D.

*Figure 11.* The LLM embedding space of MAGELLAN displayed using t-SNE with goals used in Q2 (Train) and Q3 (Test), along with MAGELLAN's estimated success probability and linear interpolation between goals. We show the embedding space for a single seed for the four architectures described in Appendix D.1 at the end of the 500k training episodes.

## D.2. Q1. Competence estimation properties

### D.2.1. PER-GOAL COMPETENCE ESTIMATION

In this section, we report the evolution of the per-goal competence estimation of experiments from Section 4.1. We show it separately for the three goal space size: 25k (Figure 12), 50k (Figure 13) and 100k (Figure 14).

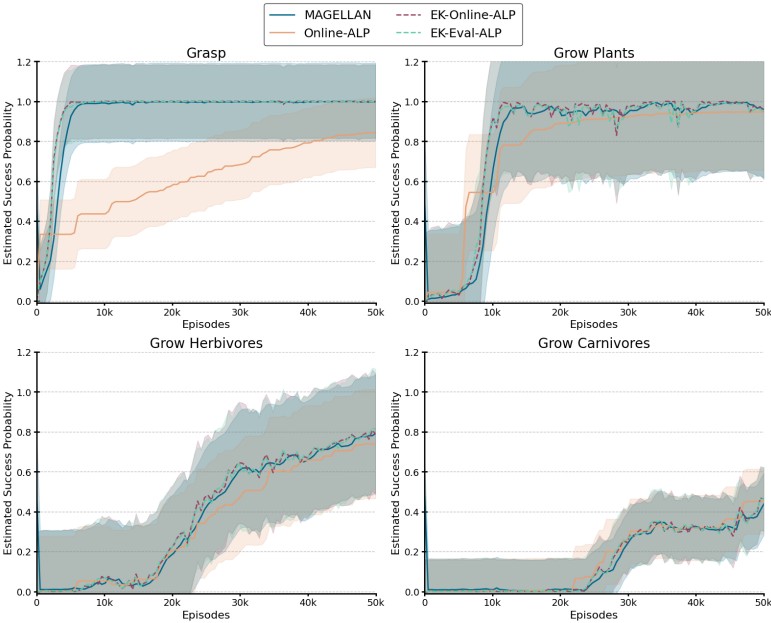

*Figure 12.* Evolution of competence estimation for each ALP method on each goal category for 25k goals. We show the average competence and its standard deviation across 8 seeds that use EK-Eval-ALP to sample goals.

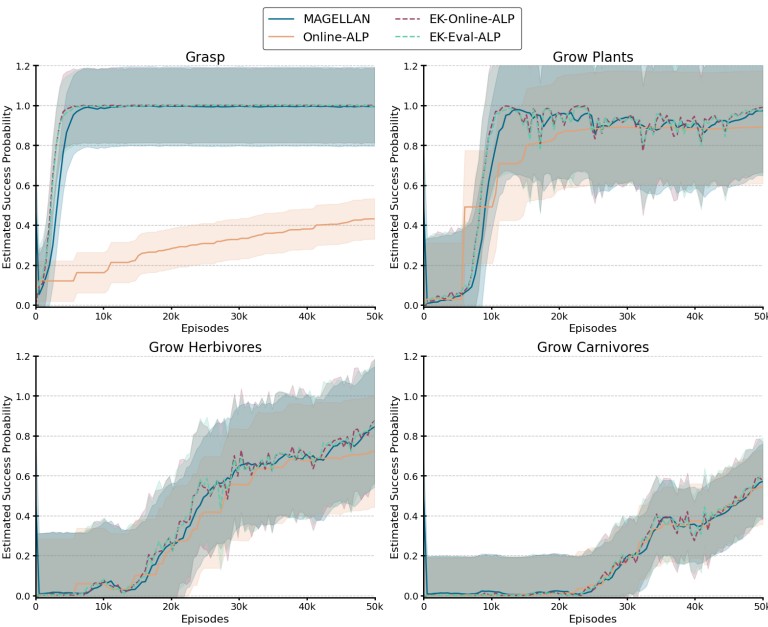

*Figure 13.* Evolution of competence estimation for each ALP method on each goal category for 50k goals. We show the average competence and its standard deviation across 8 seeds that use EK-Eval-ALP to sample goals.

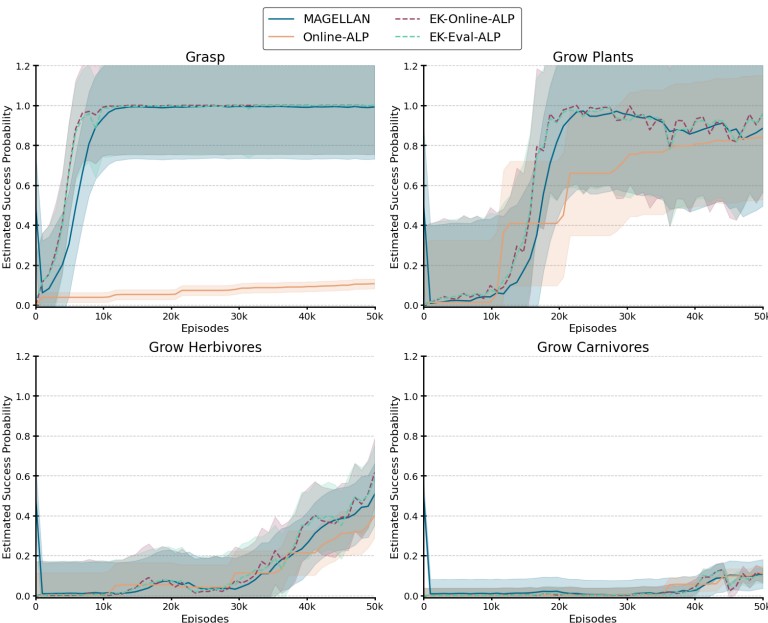

*Figure 14.* Evolution of competence estimation for each ALP method on each goal category for 100k goals. We show the average competence and its standard deviation across 8 seeds that use EK-Eval-ALP to sample goals.

### D.2.2. COMPETENCE ESTIMATION ON THE BABYAI-TEXT GOAL-SPACE

We replicated the experiment from Section 4.1, simulating a learning agent and estimating its competence online using MAGELLAN and Online-ALP. In this setting, the goal space is drawn from the BabyAI-Text environment, consisting of five goal categories of increasing difficulty: Go to *"object"*, Pick up *"object"*, Open *"door"*, Put *"object A"* next to *"object B"*, and Pick up *"object A"* then go to *"object B"*. As shown in Figure 15, MAGELLAN outperforms Online-ALP, successfully tracking the agent's competence progression across the different task categories.

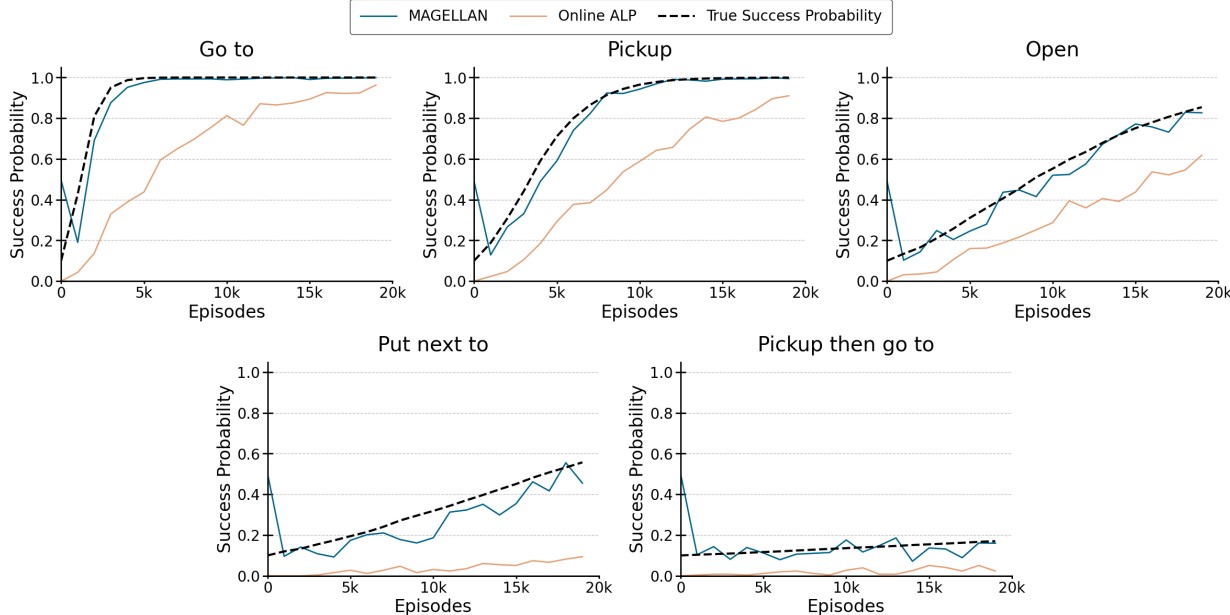

*Figure 15.* Competence estimation on BabyAI-Text. MAGELLAN accurately tracks competence across five goal types of increasing difficulty and consistently outperforms Online-ALP.

### D.2.3. IMPACT OF THE LLM USED IN MAGELLAN

In this section, we compare different LLMs (Flan-T5-small, Flan-T5-base, and Qwen2.5-0.5B (Qwen et al., 2025)) to evaluate their impact on the competence estimation performance of MAGELLAN. The experiments simulate agent learning using two goal spaces: OpenR1-Math-220k and BabyAI-Text (Figures 16 and 17). Results show that all tested LLMs achieve similar performance, suggesting that MAGELLAN's competence estimation remains robust across a range of lightweight language models.

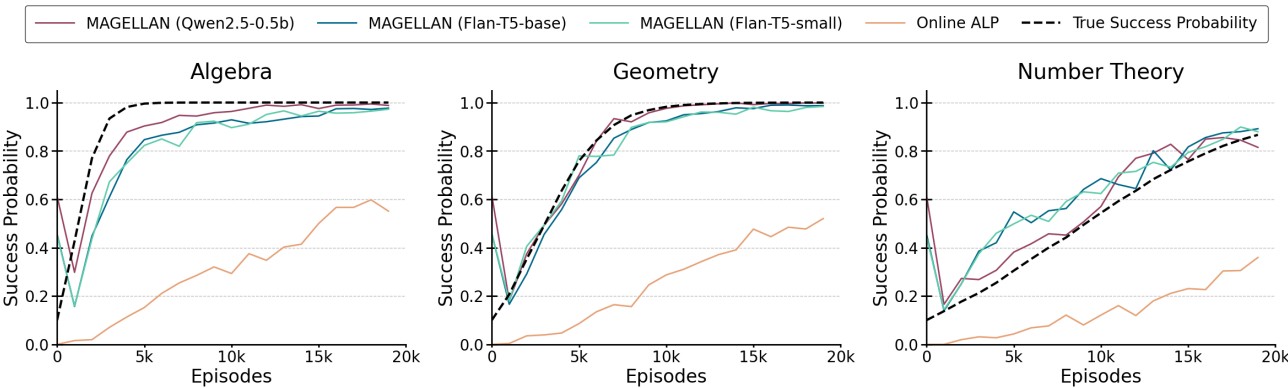

*Figure 16.* Effect of LLM choice on MAGELLAN's competence estimation for OpenR1-Math-220k. All models yield similar performance.

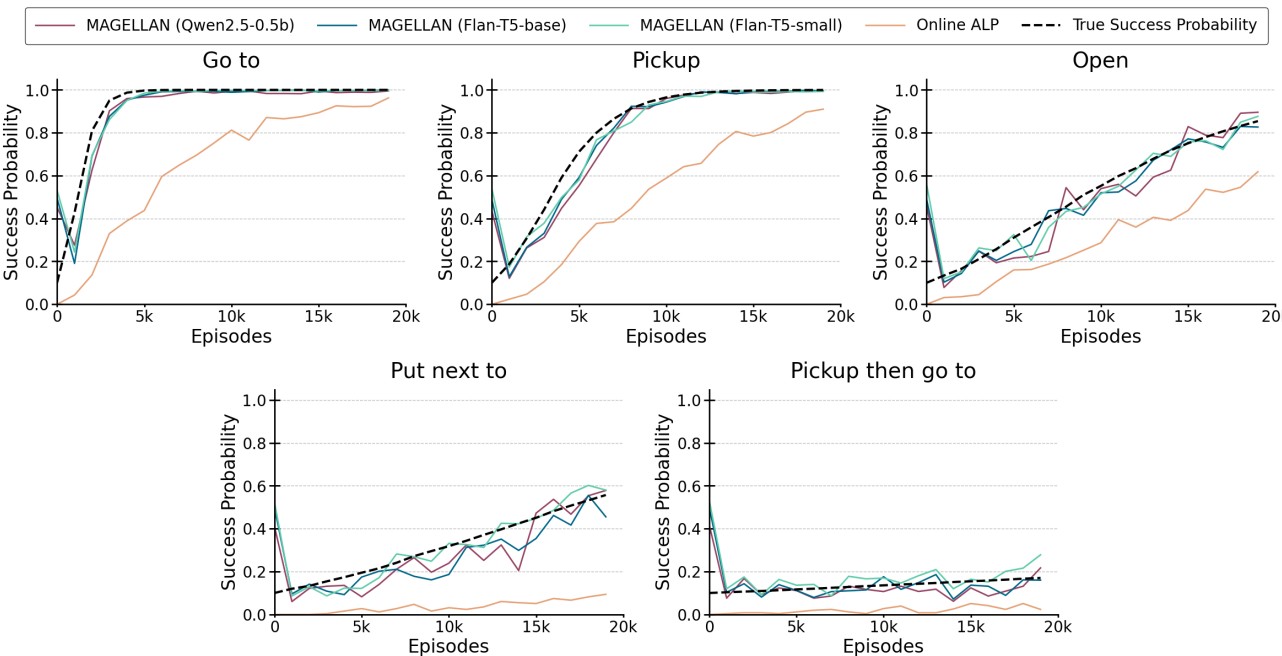

*Figure 17.* Effect of LLM choice on MAGELLAN's competence estimation for BabyAI-Text. Performance remains consistent across models.

## D.3. Q2. Training an LLM-based RL agent with MAGELLAN

### D.3.1. PER-GOAL SUCCESS RATE

We detail in Figure 18 the evolution of the success rate per method and per goal type. All methods perform similarly on the type "Grasp". However, MAGELLAN outperforms all the baselines that do not rely on expert knowledge (Uniform and Online-ALP) on all other types. Learning dynamics fostered by MAGELLAN are close to the ones generated with EK-Online-ALP which relies on expert knowledge. Such results hint towards a clustering of the goal space by MAGELLAN to efficiently and reliably estimate the SR of the LLM agent. This hypothesis is strengthened by the plot of the embedding space in Appendix D.4.

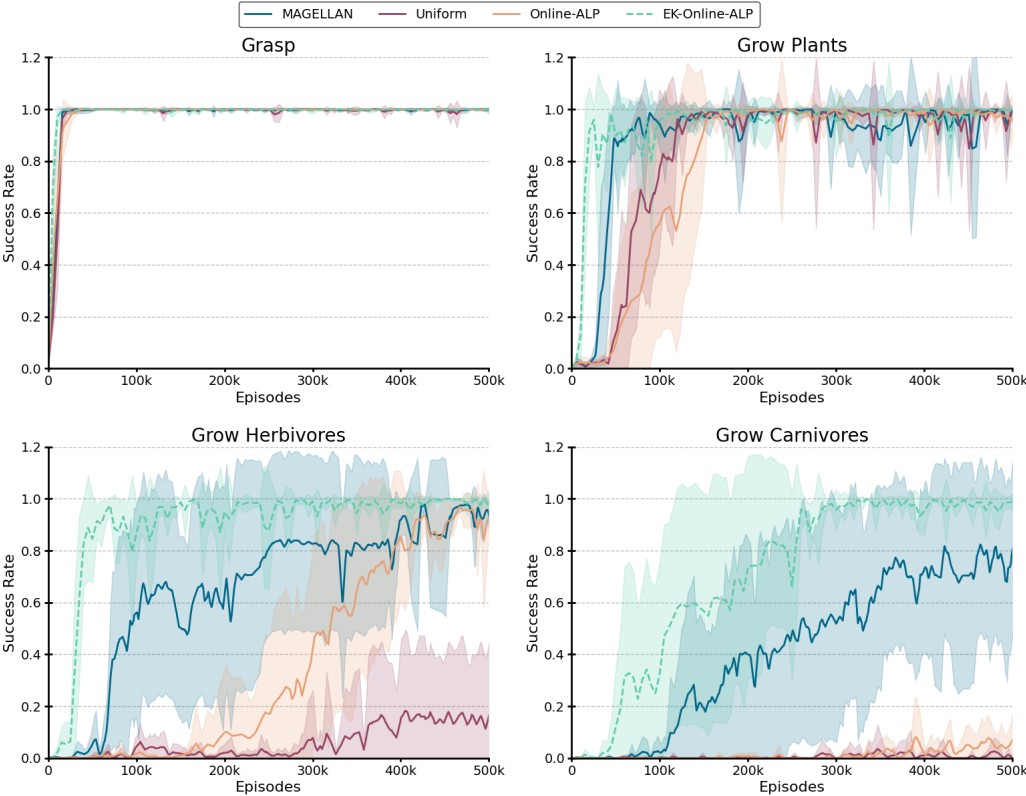

*Figure 18.* Evolution of average SR for each ALP method for each goal category. To get the success rate within a category, we evaluate the policy on 64 goals of this category. We use 8 seeds to plot the mean and the standard deviation.

### D.3.2. GOAL SAMPLING STRATEGIES

We analyze the sampling strategies of the different methods. In Figure 19, it appears that MAGELLAN has a sampling strategy close to the one of EK-Online-ALP. The main difference is that EK-Online-ALP does not select impossible goals thanks to the use of expert knowledge. MAGELLAN's sampling strategy quickly begins to sample goals of type "Grow herbivore" after sampling goals of type "Grow plant", which contrasts with the strategy of Online-ALP. Indeed, Online-ALP mostly selects goals already encountered, getting stuck on one goal type before moving to another. The similarity between MAGELLAN and EK-Online-ALP indicates that MAGELLAN is able to cluster the goal space dynamically, using this information to generalize its competence estimation to sample interesting goals not yet discovered. More results on the dynamic clustering of the goal space by MAGELLAN are given in Appendix D.4.2.

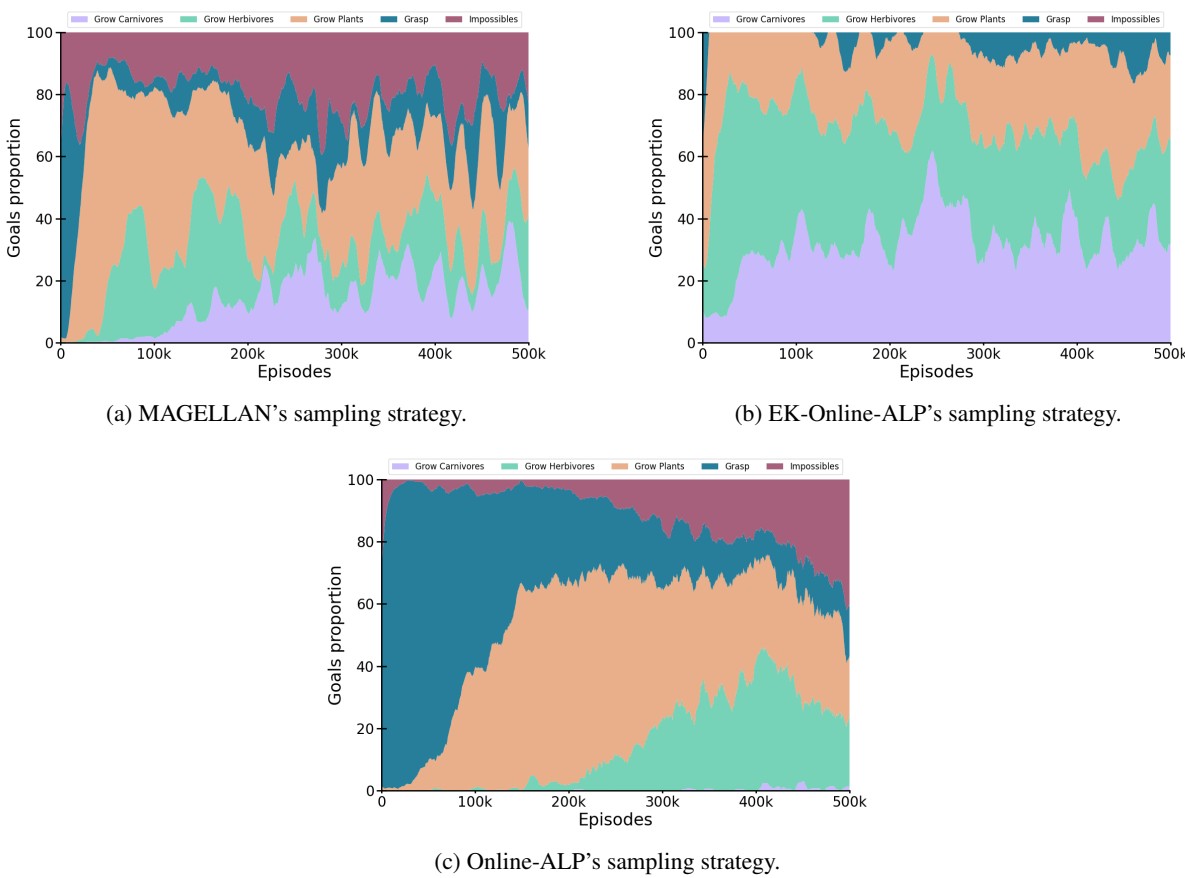

(a) MAGELLAN's sampling strategy.

(b) EK-Online-ALP's sampling strategy.

(c) Online-ALP's sampling strategy.

*Figure 19.* Goal sampling strategies of MAGELLAN, EK-Online-ALP, Online-ALP. We do not take into account the 20% uniformly sampled goals from the $\varepsilon$-greedy exploration.

## D.4. Q3. MAGELLAN's generalization abilities

### D.4.1. PER-GOAL SUCCESS PROBABILITY ESTIMATION ON TEST SET

In this section, we provide a detailed analysis of the results presented in Section 4.3, specifically examining the ability to generalize success probability estimations to test goals across different goal types (see Figure 20). As expected, Online-ALP exhibits the largest errors, as it can only assign a success probability of 0 to unseen goals. In contrast, both MAGELLAN and EK-Online-ALP achieve highly accurate estimations for "Grasp", "Grow plant", and "Grow herbivore". However, both methods tend to overestimate the generalization abilities of the policy on "Grow carnivore" goals.

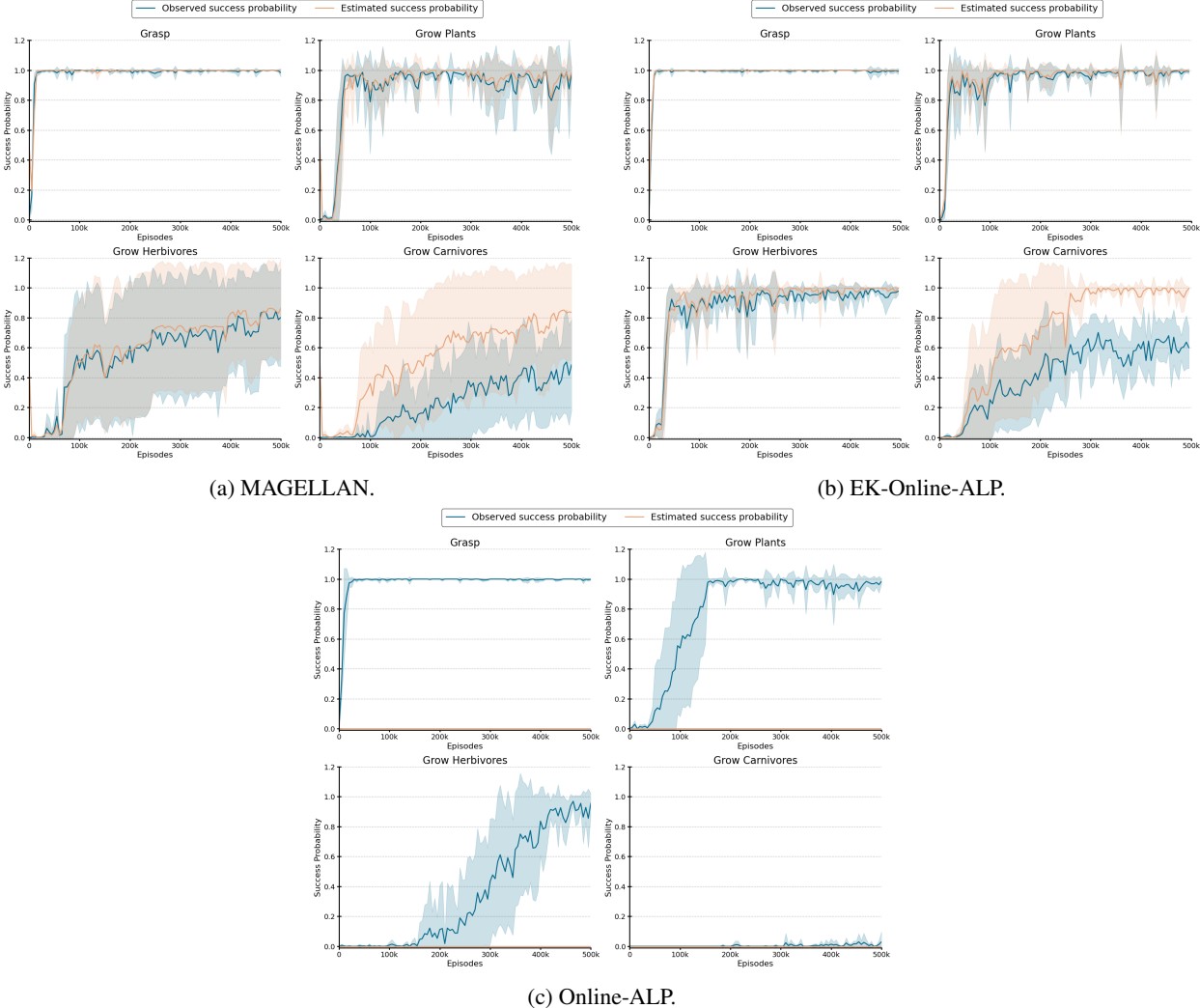

*Figure 20.* Per-goal estimation of the success probability for each method. The average result (over 8 seed) is the solid line and the shaded zone represents the standard deviation.

D.4.2. EVOLUTION OF THE EMBEDDINGS

In Section 4.3, we present the embedding at the beginning and the end of the training for the seed 0.[3] In this section, we explore how the goal space is dynamically modified throughout training with Figure 21. It is a chronogram of the embedding space used by MAGELLAN projected using t-SNE (van der Maaten & Hinton, 2008) with the estimated success probability over the whole space calculated using linear interpolation with a Gaussian filter. Each embedding (except the first one that is the initial state) is plotted after the agent masters a goal category.

- At the beginning, Figure 21a, no structure is discernible in the goal space and the estimated success probability is uniform around 0.5.

- After mastering the goals of type "Grasp", Figure 21b shows several clustered appeared. All "Grasp" goals are in the same cluster with a high estimated success probability zone. The goals of type "Grow plant" are also clustered together and close to the zone of high estimated success probability. That is a hint that MAGELLAN has already picked them as the next candidate type for the curriculum. It also correctly places the "Grow plant" goals from the test set in the same cluster as the ones from the train set. However, it still mixes them with impossible goals, underlying that it does not fully master this type of goal. The "Grow herbivore" and "Grow carnivore" are mixed with other impossible goals.

- After achieving mastery of "Grow plant", in Figure 21c, we see both "Grasp" and "Grow plant" are correctly clustered in the zone of high success probability, with the goals from the test set correctly placed into the two clusters. The goals from the type "Grow herbivore" and "Grow carnivore" are clustered together and almost separated from the impossible goals. Their cluster is close to the zone of high estimated probability.

- Once "Grow herbivore" has been solved, we see in Figure 21d that the three achieved goal types are clustered in three groups, all in the zone of high estimated success probability. The "Grow carnivore" are far from this zone but still clustered together, apart from the impossible ones. It also appears that the model places "Grow carnivore" goals from the test close to the "Grow herbivore".

- Finally, when the type "Grow carnivore" is mastered, in Figure 21e, all the goals from the possible goal types (from both train and test) are placed in the zone of high success probability. The goals of type "Grow X" form three clusters tightly packed. The "Grasp" type is separated from the other types.

What is clear from the chronogram is that MAGELLAN's learning modifies the embedding space in a way that facilitates prediction of the probability of success.

---

[3]The dynamic is the same for the other seeds.

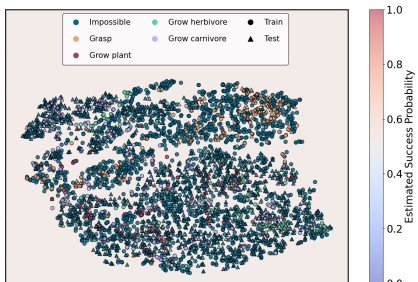

(a) Beginning of training, no goal mastered.

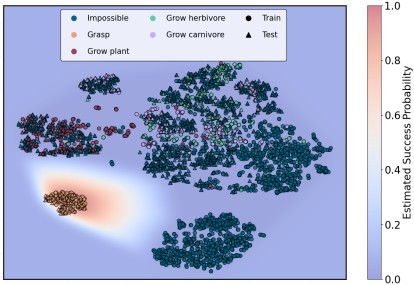

(b) "Grasp" type of goals mastered.

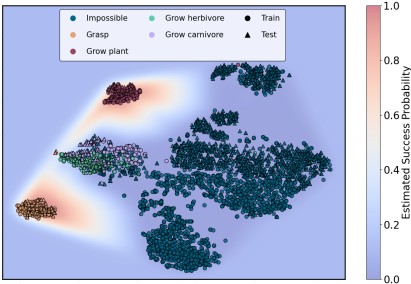

(c) "Grow plant" type of goals mastered.

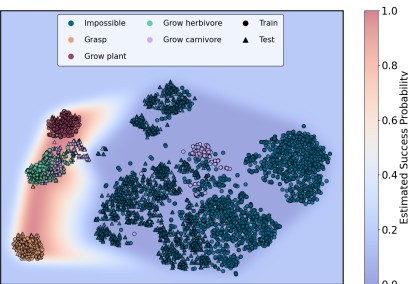

(d) "Grow herbivore" type of goals mastered.

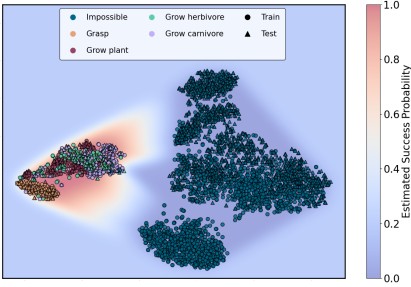

(e) "Grow carnivore" type of goals mastered.

*Figure 21.* Chronogram of the embedding space of the seed 0, at the beginning and after mastering each type of goal.

### D.4.3. EMBEDDING OF IMPOSSIBLE GOALS

As detailed in Section A.3, a goal is considered impossible either due to the absence of a required element or when attempting to grow furniture. Accurately identifying impossible goals and generalizing this knowledge is crucial for strong performance in the Little-Zoo environment. In this section, we analyze how MAGELLAN handles impossible goals.

Figure 22a illustrates the clustering of different categories of impossible goals. The impossible "Grasp" goals form a compact cluster, while the "Grow" goals—categorized as "Grow plant", "Grow herbivore", "Grow carnivore", and "Grow furniture"—exhibit four less-defined clusters. Additionally, a large, mixed cluster contains various impossible "Grow" goals. However, when examining the same embeddings through the lens of missing elements, as shown in Figure 22, a distinct structure emerges: MAGELLAN also organizes goals based on the absent element in the scene.

For instance, the previously observed clusters of impossible "Grow" goals largely align with the "missing water" category, as water is a prerequisite for all "Grow" goals. Moreover, the central mixed cluster from Figure 22a is further clustered into sub-clusters based on the missing element. In particular, the red cluster at the bottom of Figure 22 represents the absence of a plant, encompassing both "Grow herbivore" and "Grow carnivore" goals. This suggests that MAGELLAN effectively captures underlying environmental dependencies to determine goal feasibility. This ability to infer structural properties of the environment likely contributes to the strong generalization performance observed in Section 4.3 and Section 4.4.

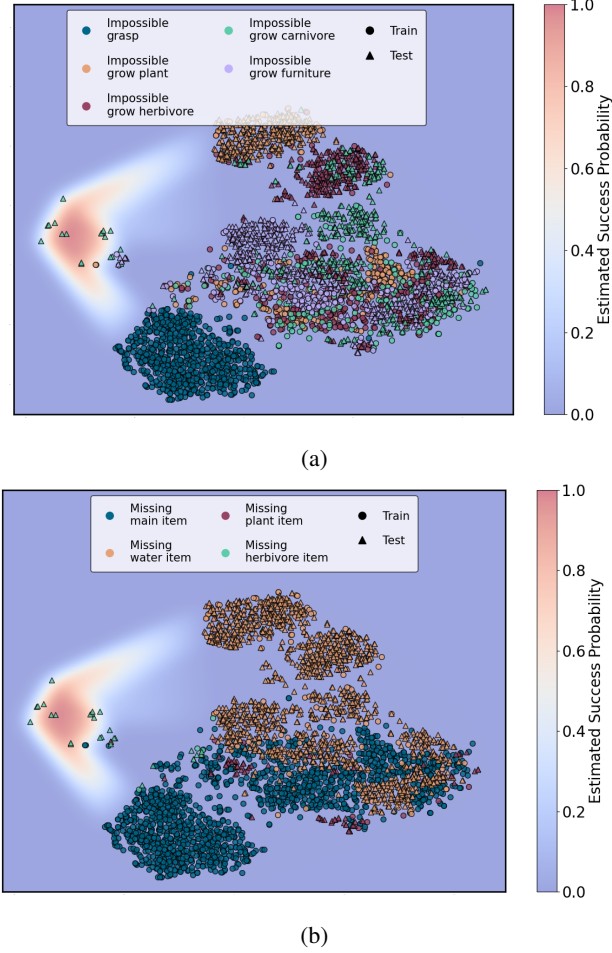

(a)

(b)

*Figure 22.* Plot of the t-SNE-projected embeddings of impossible goals, where each goal is color-coded based (a) on the type of impossible goal (b) on the specific missing element required to make it possible.

## D.5. Q4. Leveraging the generalization abilities when facing new goals

### D.5.1. 10 ADAPTATION CASES THROUGHOUT TRAINING

In Section 4.4, we present adaptation tests where the goal space evolves by replacing goals with new unseen goals from similar categories. We conduct the test by training an agent using MAGELLAN for 500k steps. Then, every $50000 * n$ episodes (with $n \in \{1; 2; 3; 4; 5; 6; 7; 8; 9; 10\}$), we stop training and replace MAGELLAN by one of the four ALP methods of goal sampling (MAGELLAN, EK-Online-ALP, Online-ALP, and Uniform). Finally, we resume the training for 50k steps and measure the SR. For EK-Online-ALP, we make it track the agent's competence based on goals sampled by MAGELLAN during the first phase, allowing it to start with an estimation on the new unseen goals. Our test measures the ability of a method to estimate and quickly update competence on unseen goals. Figure 23 details all the results obtained at the ten points we use in the experiments.

We can divide the different experiments in 2 typical scenarios:

- **Scenario zero LP (Figure 23a):** the agent has mastered the "Grasp" and "Grow plant" goals and has 0 ALP across all goals. In this scenario, all ALP estimations are equivalent. EK-Online-LP manages to discover new ALP niches faster as all impossible goals are in the same group.

- **Scenario with LP**:

  - **High LP (figures 23b, 23c, 23d):** the agent is getting a high ALP as it is learning some "Grow carnivore" goals. Here, MAGELLAN outperforms baselines by generalizing its ALP estimation and continuing training on these goals.
  - **Medium LP at the end of training (figures 23e to 23j):** the agent has almost learned all the goals, and it only remains few goals in the "Grow carnivore" category on which its performance is not stable. When changing the goal space, Online-ALP and Uniform take a lot of steps to find the remaining goals with LP in the new goal space, destabilizing the agent. As a result, the agent's SR decreases just after the transition and does not recover after 50k training epispdes. MAGELLAN and EK-Online-ALP maintain their high SR.

  We notice that the learning curve of the agents using MAGELLAN and EK-Online-ALP are almost identical, but our method does not rely on expert knowledge and naturally clustered the goal space as seen in Figure 6b.

### D.5.2. GLOBAL SAMPLE EFFICIENCY ASSESSMENT

To obtain a quantitative analysis of the results from Section 4.4, we plot in Figure 24 the average sample efficiency (after $\kappa$ episodes) of each method, averaged over the 10 tests, with $\kappa \in \{10000; 20000; 30000; 40000; 50000\}$. The sample efficiency after $\kappa$ episodes is calculated as $\frac{1}{10} \sum_{n=1}^{10} \int_0^\kappa SR(50000 * n + t) - SR(50000 * n) \, dt$.

We observe that the Uniform method is the only approach exhibiting a negative average sample efficiency, which is expected since it predominantly samples impossible goals, thereby destabilizing the agent. In contrast, the other methods demonstrate increasing sample efficiency as $\kappa$ progresses, reflecting improved success probability estimation and, consequently, more effective goal sampling. However, Online-ALP exhibits only a marginal improvement, as it must continually re-estimate success probabilities across the entire goal space.

Both MAGELLAN and EK-Online-ALP, which leverage generalization to estimate success probabilities for novel goals, achieve higher sample efficiency. Notably, EK-Online-ALP benefits significantly from expert knowledge in clustering the goal space and identifying impossible goals. However, its strong performance is contingent on the assumption that new goals belong to the same categories as those in the training set, which may limit its generalization capacity.

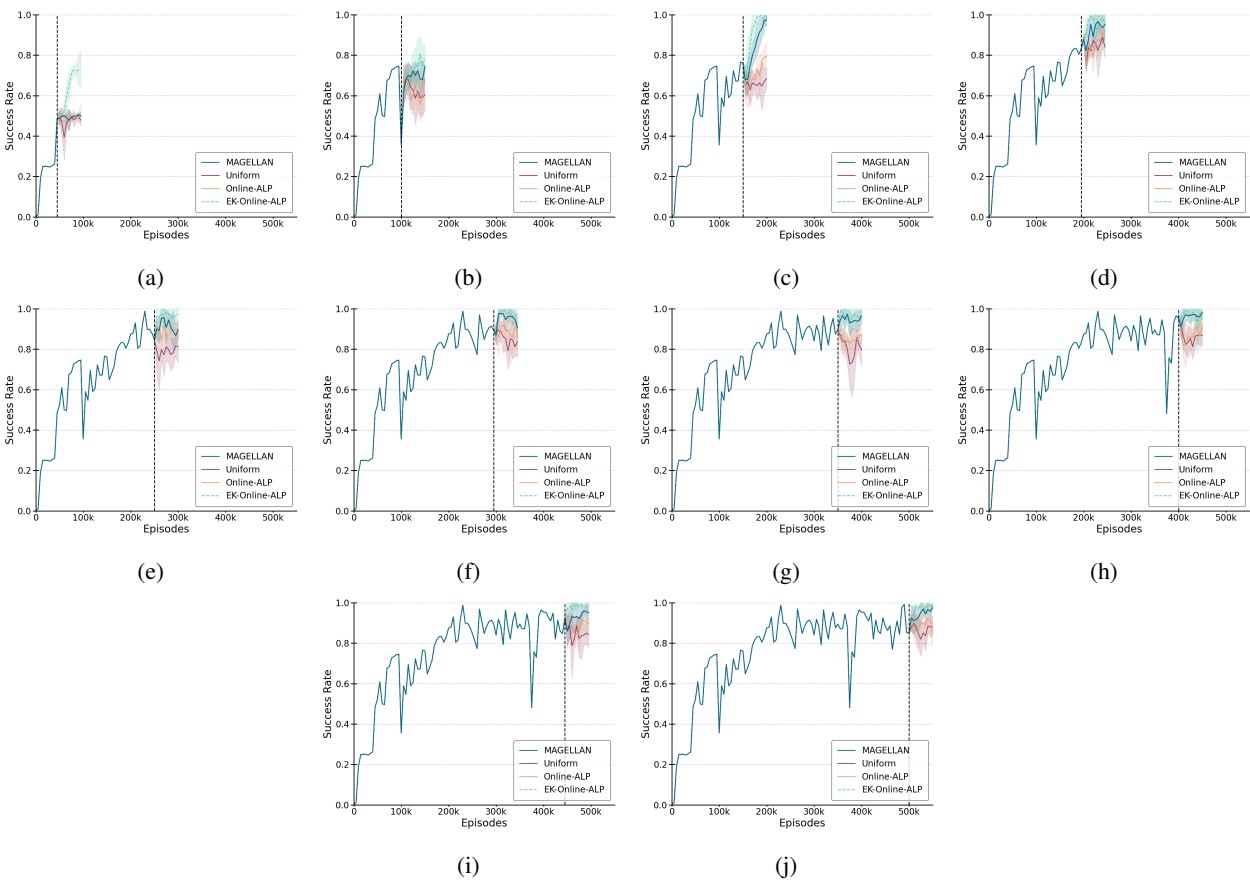

*Figure 23.* **Adaptation tests**: Using a single's seed training of 500k episodes, we stop and replace all goals by new unseen goals every 50k episodes. We then resume training by sampling goals using each of our four methods' ALP estimation (MAGELLAN, EK-Online-ALP, Online-ALP, Uniform) and perform 50k training episodes. We report the evolution of observed competence (SR) when evaluating the policies on 64 goals per category from the new training set every 5000 episodes. We report the average competence over evaluated goals along with standard deviation.

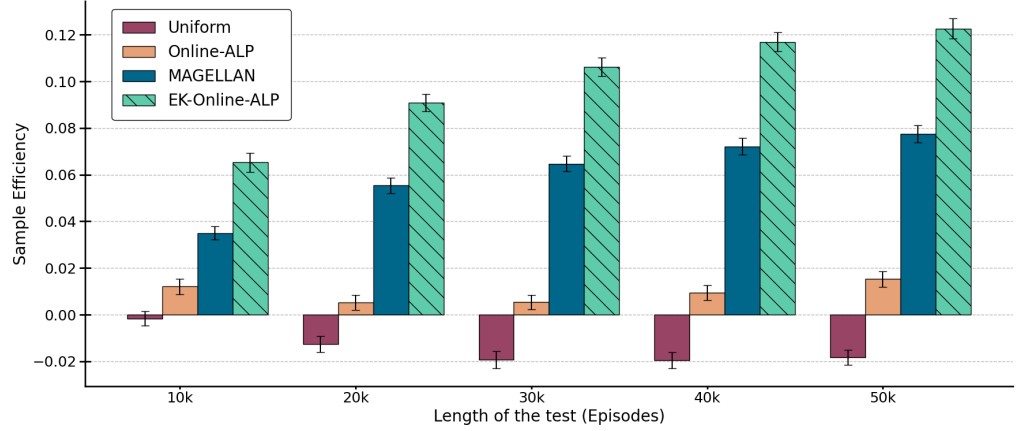

*Figure 24.* Average sample efficiency (after $\kappa$, the length of the test) of each method average over the 10 tests.

