# OpenReview forum: "MAGELLAN: Metacognitive predictions of learning progress guide autotelic LLM agents in large goal spaces"
_ICML.cc/2025/Conference — ICML 2025 poster_

### Official Review · Reviewer_pnae · 2025-02-24

**Overall Recommendation:** 1

**Summary:**

This paper tackles the problem of learning LLM agents in large goal spaces. This paper considers the situation where an LLM maximizes the expected success probability over the huge number of goals. To train the LLM more efficiently, the paper proposes a goal selector that chooses the best goal to achieve based on a neural competence estimator based on previously collected trajectories. The proposed method demonstrates better generalizability over the goal space compared to other baselines.

**Claims And Evidence:**

The main claim of the paper is that a neural-based competence estimator trained on small subset of goals seen during training can be generalized to unseen goals. I believe that this claim is well supported by Table 2 and Figure 5.

**Essential References Not Discussed:**

To me, this work is closely related to unsupervised environment design (UED), where a neural network generates novel environment instances to facilitate agent learning [1, 2]. If each goal is treated as a separate MDP, the setting in this paper is identical to UED. I recommend discussing the relationship between the proposed method and UED to provide clearer context. I have introduced only the most well-known paper on UED, but numerous related papers have been published over the past five years.

[1] Dennis et al., Emergent Complexity and Zero-shot Transfer via Unsupervised Environment Design, NeurIPS 2020 \
[2] Jiang et al., Replay-Guided Adversarial Environment Design, NeurIPS 2021

**Experimental Designs Or Analyses:**

Overall, I find the experimental design and analysis well conducted. However, I have some concerns about the simplicity of the goal format and would like to see an analysis on the failure modes of the proposed method.

**Methods And Evaluation Criteria:**

I find the proposed method reasonable. However, I have concerns regarding the significance of the performance improvement and the simplicity of the baselines considered. Specifically, the observed improvement is not entirely surprising, as online-ALP is naturally expected to exploit previously seen goals with high competence scores, given that there is no way to estimate competence for unseen goals. The only difference is the introduction of a neural network to enhance generalizability.

Additionally, I have concerns about the presentation of the proposed benchmark. Since it is newly introduced, I believe the authors should dedicate more space in the main text to explaining it in detail. At a minimum, they should provide examples of goals or describe their structure, as goal structure plays a crucial role in generalization. From my understanding, all goals follow a simple format, such as "grasp something" or "grow something," which makes them relatively easy for an LLM to generalize.

**Other Comments Or Suggestions:**

N/A

**Other Strengths And Weaknesses:**

The writing and presentation were clear to me, except for the explanation of the proposed benchmark.

**Questions For Authors:**

Please refer to "Methods And Evaluation Criteria" for my specific concerns.

**Relation To Broader Scientific Literature:**

N/A

**Theoretical Claims:**

N/A

---

> ### Author Rebuttal · Authors · 2025-04-01
>
> We thank reviewer pnae for their detailed reading of our manuscript, finding our method reasonable and our experiments well-designed and supporting our conclusions. We now address the reviewer’s concerns.
>
> ## Novelty / contribution concerns
> We acknowledge the reviewer’s concerns regarding the simplicity of our baselines and the contribution of our approach.
>
> First, the reviewer rightfully pointed out that Online-ALP cannot estimate competence over unseen goals. However, we would also like to remind the reviewer that our literature review on LP estimators in Section 2.2 and Appendix B highlights that: 1) to this day, the baselines we implemented in our paper (including Online-ALP) are **the only approaches that exist for estimating LP over discrete goal spaces** and 2) the baselines based on periodic evaluation of the competence over the full goal space are intractable in large goal spaces.
>
> Second, the reviewer mentions that our main contribution is the introduction of a parametric LP estimator able to generalize from seen goals to unseen ones. However, while few prior works studied the use of neural networks to learn latent representations of continuous goals, our work is the first to study parametric LP estimators and their generalization abilities for natural language goals (which have become ubiquitous in the era of LLM agents). Moreover, Appendix D.1 shows that learning such an estimator is not trivial. In particular, we show that learning a neural network on top of a fixed and pretrained embedding model totally fails to build semantic relationships useful for precise LP estimation. Our experiments show that MAGELLAN successfully estimates and generalizes LP by leveraging and adapting the embedding abilities of pretrained LLMs.
>
> ## Explanations on Little-Zoo
> We acknowledge that the section explaining Little-Zoo in the main paper (Section 3.4) is fairly short, with details left to Appendix A. As the reviewer suggested, we propose to improve this section, especially to explain how goals are constructed. **In particular, the reviewer’s comment made us realize our explanation may be confusing as it led to a misunderstanding of two fundamental properties of our environment**:
> **1)  goals are not an instruction alone but rather the combination of an instruction and the description of a scene initialization (i.e. objects accessible)**. For instance, here is a feasible goal: “Goal: Grow deer. You see: baby deer, bookshelf, water, tomato seed.”. And here is an impossible one (water is needed to grow the seed): “Goal: Grow deer. You see: baby deer, bookshelf, baby lion, tomato seed.”.
> **2) Little-Zoo has multiple categories of objects (i.e. plants, herbivores and carnivores) and each category requires a specific sequence of actions to grow one of its objects.**
>
> Consequently, estimating (and generalizing) well an agent’s LP in Little-Zoo not only requires one to understand the inner families of objects but also necessitates one to discern what makes an instruction possible or not in a given scene (i.e. discovering the tech tree).
>
> We give all these details in Appendix B, including the optimal actions sequence to solve the possible instructions, but we will update Section 3.4 so that Little-Zoo’s natural complexity is easier to grasp.
>
> ## Essential References Not Discussed
> We thank the reviewer for identifying references not discussed. In particular, UED approaches are **Automatic Curriculum Learning (ACL) methods** that do not assume a discrete set of goals to sample from and that, to the best of our knowledge, do not make use of Learning Progress. In Section 2.1, we discuss various ACL methods but do not provide an extensive review of them as our paper does not introduce yet a new ACL approach.
>
> Indeed, we focus on 1) improving the estimation of **Learning Progress on natural language goal spaces** and 2) augmenting LLM agents with a metacognitive monitoring skill.
>
> One can use this LP estimation (as we have done in Section 4.2) to scaffold an RL learner’s curriculum using an ACL method but we did not propose any contribution to the latter. As explained in Section 3.2, we reused for this prior approaches leveraging a Multi-Armed Bandit framework to sample goals according to their LP estimation.
>
> We thus propose to add UED references to the ACL methods mentioned in Section 2.1.

---

> > ### Comment · Reviewer_pnae · 2025-04-02
> >
> > Thank you for your response. You mentioned that goals are not just instructions, but a combination of an instruction and a description of the scene initialization. That explanation actually makes your paper seem even more related to UED. Also, UED does make use of learning progress—you can even find the exact term "learning progress" explicitly mentioned in [1]. One major difference from prior work on UED is the use of a language model. However, I believe this alone may not be sufficient to claim strong novelty for the paper. Therefore, I maintain my score.
> >
> > [1] Dennis et al., Emergent Complexity and Zero-shot Transfer via Unsupervised Environment Design, NeurIPS 2020

---

> > > ### Author Response · Authors · 2025-04-04
> > >
> > > We thank the reviewer for their prompt reply to our rebuttal.
> > >
> > > ## Relation to UED
> > > We believe the reviewer is not representing our work and its relation to the UED literature fairly. We propose to explain why and have revised the manuscript to better convey these distinctions.
> > >
> > > While the mentioned paper does contain the words “learning progress” (only in the abstract and related works sections, without any explicit link to the proposed method): **this paper does not use Learning Progress (LP) in its methods**. Rather, it approximates a regret measure between the agent's current competence and the maximum competence reachable, which is then used to select goals with maximum regret. This regret measure is intractable and has been approximated using various strong assumptions, e.g. having access to an oracle / optimal policy or considering that all goals have the same maximum competence (hence there are no impossible goals). In comparison, the LP solely relies on the agent's competence, without any assumption on an optimal policy's competence. Furthermore, in addition to naturally dealing with impossible goals, the LP is particularly efficient when given a limited training budget in which it is not possible to learn all goals (Lopes and Oudeyer, 2012). Indeed, focusing on goals with maximum LP leads to maximizing the number of goals learned over the given budget. The regret used in UED does not have the same property, which is key as, in practice, most researchers do not have an infinite compute budget. As an example, if one had access to the maximum competence reachable in Little-Zoo, the curriculum induced by the regret would be: Grow carnivores -> Grow herbivores -> Grow plants -> Grasp (as the hardest goals are slower to learn, and thus posess a higher regret for longer than easier goals). In comparison, the LP leads to the following curriculum: Grasp -> Grow plants -> Grow herbivores -> Grow carnivores. This advantage of empirical LP estimation has already been shown in previous papers we cite, and our objective in this paper is to show how the LP approach can be extended to large discrete semantically structured goal spaces (thus we do not aim here to compare again LP approaches with other methods of ACL).
> > >
> > > ## The only contribution is the use of a Language Model
> > > Moreover, the reviewer reduces our contribution to “the use of a language model”. Again, as highlighted both in our paper and in our rebuttal, our work proposes a new LP estimator over large and discrete goal spaces. As covered in Section 2.2 of our manuscript, **it is known that estimating LP over large goal spaces is hard** (Stout and Barto, 2010; Lopes and Oudeyer, 2012; Kanitscheider et al., 2021; Zhang et al., 2024). This is notably explained by the fact that estimating LP requires tracking the current (and past) competence of the learner over the full goal space. Efficiently estimating the LP thus requires generalizing the learner’s competence from goals it has practiced to unseen goals. **To this day, no method exists to efficiently estimate LP over large discrete spaces. In our paper we propose MAGELLAN as the first instance of such an estimator**. Using an LLM inside our estimator, we show MAGELLAN is particularly efficient on natural language goal spaces. Moreover, we show in our results (Figure 5, Appendix D.1, Appendix D.4.2) that simply using an LLM is not sufficient: MAGELLAN only works if its objective is used to also finetune the LLM and adapt its representations to the environment’s semantic (and we study and visualize properties of these learned representations)
> > >
> > > ## Experiments on new domains
> > > Besides these comments, we take this occasion to provide new results related to the generality of our evidence on how MAGELLAN generalizes its competence estimation over complex goals. In our response to reviewer 3cqQ, we showed that MAGELLAN accurately estimates and generalizes a learner’s competence (and by extension LP) on various types of math problems when equipped with Qwen2.5-0.5B. **We performed additional experiments** comparing MAGELLAN’s performance with two smaller LLMs (Flan-T5 80M and 248M). These results confirm our response to reviewer 9oss: **smaller-scale LLMs can also learn to capture the semantic relationships between goals**.
> > > We also provide even new results with a simulated learner on another embodied environment (closer to our Little-Zoo): BabyAI-Text (Carta et al., 2023). We created synonyms for each goal (e.g. “Go to the red ball” is also formulated as “To win the game you need to reach the crimson sphere”) resulting in a goal space with more than 20k goals. These results provide similar evidence to the one obtained with maths problems.
> > > We provide the goals and results for both domains on [our anonymous repository](https://github.com/ghjnkmjl745678/MAGELLAN_ICML/) (in the `math_experiments` and `babyai_experiments` folders).
> > > **MAGELLAN’s efficiency thus goes beyond Little-Zoo and is not tied to Flan-T5 248M.**

---

### Official Review · Reviewer_3cqQ · 2025-03-08

**Overall Recommendation:** 3

**Summary:**

This paper presents a framework for improving competence and learning progress (LP) estimation, used for the goal section of LLM agents in very large (even infinite) evolving goal spaces. The proposed method leverages the semantic relationships between goals and an LLM’s internal semantic knowledge to improve competence prediction. The method is compared to baseline LP estimation methods that rely on empirical evaluations of goals, both online during training or offline and with or without expert knowledge. The agent is tested on a proposed environment called Little-Zoo, which is a text-based learning environment designed to assess generalization and adaptation to evolving goal spaces. Using the Little-Zoo environment, the proposed method is shown to have lower competence prediction error than all baselines that do not use expert knowledge, and goal selection based on the estimated LP leads to faster agent learning. The paper also analyzes the generalization and adaptation of the competence prediction across the goal space.

**Claims And Evidence:**

1. The proposed method predicts agent competence on goals more accurately than existing methods.

This claim is well supported in the paper, as experiments show estimation errors are much lower for the proposed method than all baselines except those that rely on expert knowledge.

2. The proposed method provides an automatic curriculum over a large evolving goal space that allows an agent to more efficiently learn to master the environment.

This claim has been shown quite convincingly for the proposed Little Zoo environment, as results show goals sampling based on LP predicted using the proposed method learns faster and to a higher final performance than all baseline methods except for one that uses expert knowledge.

However, all experiments were conducted in the Little Zoo environment. This weakens this claim since it is difficult to be confident about the generality of these results with any evidence on more environments (such as Minecraft, although it is not text-only and would require a VLM).

3. The proposed method’s competence estimation can generalize to unseen goals and adapt quickly as the goal space evolves.

The paper conducted one experiment to study generalization (section 4.3) and indeed show the proposed method has the lowest prediction error on a held-out set of tasks. Again, this is somewhat weakened as it was only conducted on Little-Zoo, which may not generalize to other environments.

**Essential References Not Discussed:**

No.

**Experimental Designs Or Analyses:**

The paper’s experiments are generally well designed and analysis sound. As mentioned above, the focus on Little-Zoo only is a limitation.

**Methods And Evaluation Criteria:**

Methods are sound. As mentioned above, the sole focus on Little-Zoo as the evaluation environment limits the confidence of the claims made in the paper. Consider evaluating the proposed method on a more diverse set of environments with large goal spaces.

Little-Zoo is design to have 80% of goals being impossible. This is quite significant. What was the motivation for this? Does this make the prediction task easier since predicting impossibility already gets to 80% accuracy?

**Other Comments Or Suggestions:**

D.3.2 Line 1382 "usinf" -> "using"

This section on the auto-curriculum over goals is very interesting.

The clustering of goals is also a very interesting illustration.

**Other Strengths And Weaknesses:**

N/A

**Questions For Authors:**

Unclear how to interpret the plus and minus symbols in Table 1, what do they mean exactly?

In Table 2, the authors note that the error in Online-ALP's Grow Herbivore is due to the policy not mastering that goal category. How do you measure this effect and how do you determine the extent other values in the table are or are not affected by this?

**Relation To Broader Scientific Literature:**

The paper is well placed within broader scientific literature, with all relevant prior works discussed in the related works section.

**Theoretical Claims:**

The paper does not make any theoretical claims.

---

> ### Author Rebuttal · Authors · 2025-04-01
>
> We thank reviewer 3cqQ for their detailed review of our paper, finding our claims well supported by the experiments, the experimental protocol well-designed, and highlighting the interestingness of our section on automatic curriculum learning and our analysis of the LLM’s embedding space. We now address the concerns raised by the reviewer.
>
> ## Single and limited environment
> We understand the reviewer’s concern about the generalizability of our approach, given that only Little-Zoo is used for evaluation. As explained in the manuscript, Little-Zoo provides a controlled setting to rigorously assess MAGELLAN’s properties. Specifically, accessing the true underlying categories of objects allows us to verify whether MAGELLAN correctly identifies them.
>
> Additionally, multiple reviewers' comments on Little-Zoo made us realize our explanations may have led to misunderstandings. Despite its name, Little-Zoo is neither trivial nor small—it consists of tens of thousands of goals. Crucially, goals are not just instructions but combinations of instructions and scene descriptions (i.e., accessible objects). **Estimating an agent’s LP in Little-Zoo requires understanding object families and discerning what makes an instruction possible in a given scene (i.e., the tech tree)**. The combination of all possible instructions and scene initializations results in 25,000 goals, 80% of which are impossible. Such a goal space, with mostly infeasible goals, also naturally arises when using freely generated natural language goals with an LLM (e.g., in an autotelic agent), as many goals wouldn’t respect the environment’s dynamics. Finally, Little-Zoo builds on prior environments like WordCraft (Jiang et al., 2020) by **introducing a key missing component: semantic relationships between goals**.
>
> Regarding the generality of our results, MAGELLAN relies on extracting structure from the semantic goal space. The LLM’s embedding space notably enables MAGELLAN to generalize (e.g., knowing which animals are herbivorous or that water is needed to grow seeds). MAGELLAN is designed for language-based goal spaces where LLM embeddings are most effective. In environments with non-language goals or sparse goal structures, the LLM’s utility is reduced, limiting generalization. In the worst case, MAGELLAN is expected to perform similarly to Online ALP.
>
> **To further support the broader applicability of MAGELLAN, especially on more complex natural language goal spaces, we conducted additional experiments**. We evaluated MAGELLAN’s ability at estimating a learner’s competence on math problems from the OpenR1-Math-220k dataset. Here, MAGELLAN must deal with highly non-trivial problems and in particular identify what is the type of each problem to generalize its competence estimation. We focus on *Algebra*, *Number theory*, and *Geometry*, leading to more than 20,000 problems. Given the limited time allowed by the rebuttal period, we did not train an RL learner but rather simulated one that has different learning dynamics on each problem type (i.e. how fast its success probability increases). Similarly to our experiments in Section 4.1, we compare the competence estimation of Online-ALP and MAGELLAN (using Qwen2.5-0.5B) when problems are sampled using an Uniform curriculum. The code can be found on our anonymous repository: https://github.com/ghjnkmjl745678/MAGELLAN_ICML/blob/main/math_experiments/.
>
> Figure *“sr_estimation_math.png”* from this repository shows that MAGELLAN accurately estimates each problem’s success probability. This indicates that it is able to find each problem’s type and generalize its competence estimation within each type. In comparison, Online-ALP leads to poor competence estimations given the large number of problems.
>
> We would like to perform the same experiment with two other LLMs (Flan-T5 80M and Flan-T5 248M) as performed in our additional experiment for reviewers 9oss’s rebuttal. Unfortunately, our cluster is currently under maintenance but we should be able to provide these results within the coming days. **If the reviewers kindly accept to respond to our rebuttal, we will provide these results**.
>
> ## Table 1
> We agree with reviewer 3cqQ that Table 1 (and Table 4) need more explanations. We proposed changes and explanations to add to the manuscript in our rebuttal to reviewer  s63G.
>
> ## Generalization abilities
> While Table 2 only shows the average error over several evaluations performed throughout training, Appendix D.4.1 and in particular Figure 16 shows the evolution of the error on the held-out test set. One can see that both the observed and estimated success probability of Online-ALP on “Grow Carnivore” goals is near 0. One can also see that this is the only case where this phenomenon happens (i.e. the observed success probability is greater than 0 on all the other plots). We therefore can affirm that this effect only affects the generalization results of Online-ALP on the “Grow carnivore” category.

---

> > ### Comment · Reviewer_3cqQ · 2025-04-08
> >
> > I thank the authors for the rebuttal and for clarifying my concerns. I maintain my original score.

---

> > > ### Author Response · Authors · 2025-04-08
> > >
> > > We thank the reviewer for their answer and are happy our rebuttal clarified their concerns.
> > >
> > > As stated in our rebuttal, we could not show all the new results. Following our cluster’s maintenance, we performed additional experiments **comparing MAGELLAN’s performance with two smaller LLMs (Flan-T5 80M and 248M) on the maths problems**. The results can be found [here](https://github.com/ghjnkmjl745678/MAGELLAN_ICML/blob/main/math_experiments/comparison_estimation_math.png) and confirm our response to reviewer 9oss: smaller-scale LLMs can also learn to capture the semantic relationships between goals.
> > > We also **performed new experiment in another domain** with a simulated learner using another embodied environment (closer to our Little-Zoo): BabyAI-Text (Carta et al., 2023). We created synonyms for each goal (e.g. “Go to the red ball” is also formulated as “To win the game you need to reach the crimson sphere”) resulting in a goal space with more than 20k goals. These results provide similar evidence to the one obtained with maths problems.
> > > We provide the goals and results on [our anonymous repository](https://github.com/ghjnkmjl745678/MAGELLAN_ICML/tree/main/babyai_experiments).
> > > **These new results show that MAGELLAN’s efficiency goes beyond Little-Zoo and is not tied to Flan-T5 248M.**
> > >
> > > **We hope these new results fully address the reviewer’s remaining concerns and kindly request that they consider increasing their score in light of these improvements.**

---

### Official Review · Reviewer_s63G · 2025-03-13

**Overall Recommendation:** 4

**Summary:**

A key challenge in learning progress prediction is modeling one’s own competence in a computationally feasible and generalizable way. The paper introduces MAGELLAN, a metacognitive framework that enables LLM agents to learn to predict their competence and LP online. MAGELLAN captures semantic relationships between goals, allowing for sample-efficient LP estimation and dynamic adaptation to evolving goal spaces.

**Claims And Evidence:**

Yes, the claims are supported with clear and convincing evidence, and experiment results that support them.

**Essential References Not Discussed:**

I don’t have any additional ones to suggest.

**Experimental Designs Or Analyses:**

Yes. The claims are clearly stated in the introduction, and each claim is soundly supported with empirical evidence.

**Methods And Evaluation Criteria:**

Yes, the paper compares MAGELLAN against several suitable baselines and does sufficient analysis to support the claims.

**Other Comments Or Suggestions:**

- line 263, “we compare MAGELLAN to the classic approaches presented in 3.3”, missing “Section”
- Figure 4, it will be helpful to have a legend for the task icons, including an indication of which tasks have to be done before others.
- It will be interesting to see how the tasks embedding on the t-SNE plot changed across the whole training, instead of just seeing the before and after plots (Figure 5).

**Other Strengths And Weaknesses:**

Strengths:
- The paper is very well-written, with thorough experiments to support each claim.
- The proposed methodology is novel and is a smart way to combine training LLM agents with RL while simultaneously estimating learning progress.

Weaknesses:
- Table 1, what is the unit or measurement of how many “+” are given?
- line 266, “These baselines would totally fail if given all goals from the same hidden family, regardless of their feasibility”. It would be helpful to provide additional clarification on why this failure occurs.

**Questions For Authors:**

1. How do you think that this approach can be generalized to other text-based environments (e.g., nethack, which has much sparser reward signals)?
2. Learning progress can also be roughly estimated by telling an LLM the agent’s current capabilities, and directly asking for what the next learnable ones are. What are some ways that these approaches can be compared with MAGELLAN?

**Relation To Broader Scientific Literature:**

The paper’s key contributions align with and extend prior work on open-ended learning, intrinsic motivation, and curriculum learning. MAGELLAN builds on existing research in LP-based goal prioritization. Unlike prior approaches that rely on expert-defined goal groupings or exhaustive evaluation, MAGELLAN leverages the generalization capabilities of LLMs to dynamically estimate competence and LP, addressing scalability challenges in high-dimensional goal spaces. The paper demonstrates how metacognitive LP estimation can enhance goal selection and learning efficiency in text-based environments.

**Theoretical Claims:**

Yes, they are correct.

---

> ### Author Rebuttal · Authors · 2025-04-01
>
> We thank reviewer s63G for their in-depth review and comments on our work, finding the claims clearly supported by our experiments, as well as highlighting the novelty of our method. We now provide answers to the comments and questions asked by the reviewer.
>
> ## Table 1
> We acknowledge that Table 1 (and by extension Table 4) currently lacks an explanation on the unit and the criteria used to assign each method’s score. Note that this was also raised by reviewer 3cqQ. We propose to update the manuscript with the following changes. First, we will replace “+++” by “high” and “ ” by “low” in the Efficiency column. Then, we will add the following explanation providing the unit and criteria:
>
> For the Efficiency property, we consider a method’s efficiency as “high” if it does not require any additional evaluation (i.e. it only uses the performance observed on goals sampled), and as “low” otherwise.
>
> For Competence transfer tracking, the “+” are given according to this evaluation:
> - absence of +: the estimated competence is updated only on sampled goals
> - +: the estimated competence is updated on a predefined goal subset the sampled goal belongs to
> - ++: the estimated competence is updated on a dynamically learned goal subset the sampled goal belongs to
> - +++: the estimated competence is updated on all goals
>
> ## Modification of the embedding space
> As the reviewer mentions, it is indeed insightful to analyze how the goal embedding space evolves over time. Such an analysis can already be seen in Appendix D.4.2. The chronogram we present there shows how the different categories are identified and how they interact. We put this analysis in appendices for now as it requires a full page.
>
> ## Failure of the baseline
> The reviewer asks for more clarification on why impossible goals are not included in the predefined categories given to baselines that use expert knowledge. These baselines are based on groups predefined in advance by human experts with a strong assumption: the goals within a group share the same learning dynamics and therefore the agent’s competence is the same over all goals in the group. If impossible goals were included, these groups would lose their relevance. Moreover, because of the large number of impossible goals, the average competence within each group will always be very close to 0. There will be no progress niche that the method can use to generate a curriculum, and performance will probably be very close to the random baseline. We will update the sentence at line 266 to make it clearer.
>
> ## Generalization to other environments
> The reviewer wonders how MAGELLAN would perform in other environments, in particular with a sparser reward signal. As a similar question was raised by most reviewers, we provide a single response, included in reviewer’s 3cqQ rebuttal. **This response notably shows new results on additional experiments launched according to reviewers’ comments.**
>
> Focusing on the reward signal sparsity property, we are not sure if the reviewer discusses the reward used by the RL policy or the signal used to train MAGELLAN and would like to remind that MAGELLAN assumes a goal-conditioned environment with a binary outcome on an episode (i.e. whether the goal has been reached or not). In Little-Zoo, the RL policy also uses a binary reward obtained on the final step only, but this is not mandatory (i.e. MAGELLAN does not make any assumption on the reward signal used by the RL policy).

---

### Official Review · Reviewer_9oss · 2025-03-22

**Overall Recommendation:** 3

**Summary:**

The paper introduces MAGELLAN—a metacognitive module that enables autotelic LLM agents to estimate their own learning progress (LP) over large, discrete, and evolving goal spaces. The approach leverages the inherent semantic understanding of an LLM to learn a goal‐conditioned competence estimator that generalizes across similar natural language goals without relying on expert-defined groupings. Experiments in a custom textual environment (Little‑Zoo) demonstrate that MAGELLAN can accurately track learning progress, generalize to unseen goals, and adapt rapidly when the goal space evolves. Overall, the paper claims that this method enables the agent to build a self-organized curriculum, leading to faster and more complete mastery compared to several baseline LP estimation techniques.

**Claims And Evidence:**

Claims: The paper claims that MAGELLAN (i) efficiently estimates LP without expensive evaluations or expert-defined groupings, (ii) generalizes competence predictions to unseen goals, and (iii) adapts to evolving goal spaces, all of which facilitate improved curriculum learning.

Evidence: These claims are supported by empirical results on the Little‑Zoo environment, where MAGELLAN is compared against baselines like Online‑ALP, Eval‑ALP, and variants that use expert-defined groupings. The experiments show lower competence estimation error, higher success rates, and faster mastery across different goal types.

**Essential References Not Discussed:**

While the paper cites many foundational works, it might benefit from discussing very recent advances in open‑ended learning and meta‑learning that do not strictly rely on LP. For instance, comparisons with recent methods leveraging uncertainty estimation or self‑supervised approaches in high-dimensional goal spaces would provide a more rounded view of the state of the art.

**Experimental Designs Or Analyses:**

Experiments are performed on the Little‑Zoo environment with varying goal space sizes and include tests for generalization (held‑out goals) and adaptation (evolving goal spaces). Multiple random seeds and thorough evaluation every set number of episodes bolster the reliability of the findings.

The paper provides detailed plots of competence estimation error, success rates, and t‑SNE visualizations of the embedding space, along with ablation studies on architectural choices.

Although the experimental design is comprehensive within the chosen setting, the use of a single, synthetic environment raises concerns about external validity. More experiments on diverse environments would help establish broader applicability.

**Methods And Evaluation Criteria:**

The approach combines an online RL framework (building on SAC‑GLAM with a finetuned Flan‑T5) with a metacognitive competence estimator that uses the LLM’s latent representations to predict success probabilities. An MLP is used on top of the LLM output, and a buffer of past model weights helps compute an absolute LP (ALP) metric. The method is evaluated using observed competence (success rate) and competence estimation error, as well as computational cost (in terms of additional evaluation episodes). The use of a custom environment designed to reflect the structure of language‑defined goals is well-motivated for the study’s aims.

The criteria and baselines chosen are appropriate for the stated problem, though reliance on a single environment limits the scope of the evaluation.

**Other Comments Or Suggestions:**

1. Consider extending experiments to additional environments or real‑world datasets to validate the generality of the approach.
2. Provide a more detailed discussion on computational overhead and scalability.
3. Address potential limitations, such as the sensitivity of performance to the choice of the underlying LLM.

**Other Strengths And Weaknesses:**

Strengths:

1. Innovative use of an LLM to dynamically learn semantic relationships among goals, eliminating the need for brittle, expert‑defined groupings.
2. Comprehensive experimental evaluation and extensive supplementary material that aid in reproducibility.

Weaknesses:

1. Limited evaluation domain: The reliance on a single synthetic environment (Little‑Zoo) raises concerns about generalizability to more complex, real‑world tasks.
2. The method’s performance with larger or more advanced LLMs is not explored.

**Questions For Authors:**

Failure Modes: Were there any observed cases where the metacognitive predictions led to suboptimal goal selection or curriculum choices? If so, how were these instances addressed or mitigated?

Generalizability: How do you expect MAGELLAN to perform in environments other than Little‑Zoo, particularly in settings with more complex or less structured natural language goals?

**Relation To Broader Scientific Literature:**

The work is well positioned within the literature on intrinsic motivation, curriculum learning, and autotelic agents.

It builds on prior work in LP estimation and goal selection (e.g., Online‑ALP, Eval‑ALP) while addressing limitations related to expert-defined groupings.

The integration of metacognitive prediction via an LLM is a notable contribution. However, additional discussion comparing this approach to alternative methods (e.g., uncertainty‑based exploration or meta‑learning techniques) would further contextualize its impact.

**Theoretical Claims:**

The paper provides a formal problem statement and introduces a competence function and ALP estimation formulation. There is no in‐depth theoretical analysis or proof (e.g., regarding convergence or sample efficiency) beyond the formulation.

---

> ### Author Rebuttal · Authors · 2025-04-01
>
> We thank reviewer 9oss for their thorough feedback, finding our approach using an LLM to estimate LP innovative, highlighting the comprehensiveness of our experimental evaluation and acknowledging the effectiveness of MAGELLAN. In the next paragraphs, we answer reviewer 9oss’ concerns.
>
> ## Single and limited environment
> We appreciate the reviewer’s feedback and understand their concerns on our limited evaluation domain. As a similar question was raised by most reviewers, we provide a single response, included in reviewer’s 3cqQ rebuttal. **This response notably shows new results on additional experiments launched according to reviewers’ comments.**
>
> ## LLM choice
> We agree with the reviewer that studying MAGELLAN’s sensitivity to various LLMs would make our evidence more robust. We thus **ran an additional experiment** for our Section 4.1 where MAGELLAN’s error on 25k goals is measured with **Flan-T5 80M and Qwen2.5-0.5B** in addition to Flan-T5 248M  (which is already in the paper). We show the results in the following plot: https://github.com/ghjnkmjl745678/MAGELLAN_ICML/blob/main/errors_littlezoo_goals.png.
>
> Our results indicate that larger models lead to slightly more precise competence estimations in Little-Zoo. However, this difference is not significative as the LLM is finetuned by MAGELLAN to adapt its embedding space for competence estimation in the given environment. We will add these results to Section 4.1.
>
> Lastly, the reviewer asks about scalability. In our experiments, we used LoRA (Hu et al., 2022), limiting the number of weights to finetune. Nevertheless, larger LLMs remain slower and more computationally consuming both at inference (i.e. to estimate the LP over the goal space and choose the next goal) and during finetuning. We will add this discussion to our conclusion.
>
> ## Other methods and baselines
> We thank the reviewer for their suggestion of additional references to discuss. Section 2.1 discusses goal selection approaches in general (including methods not using LP) and we would be happy to add these references (and the ones proposed by reviewer pnae on UED methods) in this section. Could the reviewer be more precise on which work we missed?
>
> ## Failure modes
> We thank the reviewer 9oss for their question. In our experiments, we observed no significant failure cases impacting the curriculum.
>
> However, we identify two possible failures cases resulting of bad LP estimation:
> - *The LP is overestimated*: In that case, the goal will be quickly sampled by the multi-armed bandit selector and its LP estimation will be reduced.
> - *The LP is underestimated*: In that case, we must wait for the exploration mechanism in the multi-armed bandit goal sampler to eventually sample the goal, enabling MAGELLAN to adjust its LP estimation. This is the worst case among the two.
>
> Additionally, applying MAGELLAN to a very large discrete goal space with no semantic structure would lead to poor performance (i.e. similar to Online-ALP). Moreover, we do not explicitly study the case where a goal semantic is not aligned with common knowledge internalised by the LLM (e.g. in Little Zoo, a rabbit that would act as a carnivore). However, as shown with the initial embedding space of Flan-T5 248M (Figure 5.a), MAGELLAN will progressively move it from the herbivore cluster to the carnivore one throughout training.

---

### Decision · Program_Chairs · 2025-05-01

**Decision:**

Accept (poster)

**Comment:**

This paper introduces a method that enables LLM agents to estimate their own learning progress (LP) across natural language goal spaces by leveraging the LLM's semantic understanding, which facilitates automatic curriculum learning. Reviewers appreciated the idea of leveraging an LLM's capability to estimate learning progress, eliminating the need for expert-defined goal groupings. They also found that the experiments are well designed and strong enough to demonstrate the advantage of the proposed method. However, there was a concern regarding the lack of discussion on unsupervised environment design (UED). I highly encourage the authors to clarify the connection and distinction from UED in the camera-ready version. Assuming that the authors make the suggested change, I recommend to accept this paper.